# CO-EDITBENCH: HUMAN-ALIGNED BENCHMARK FOR INSTRUCTION-BASED IMAGE EDITING WITH MULTI-DIMENSIONAL ASSESSMENT

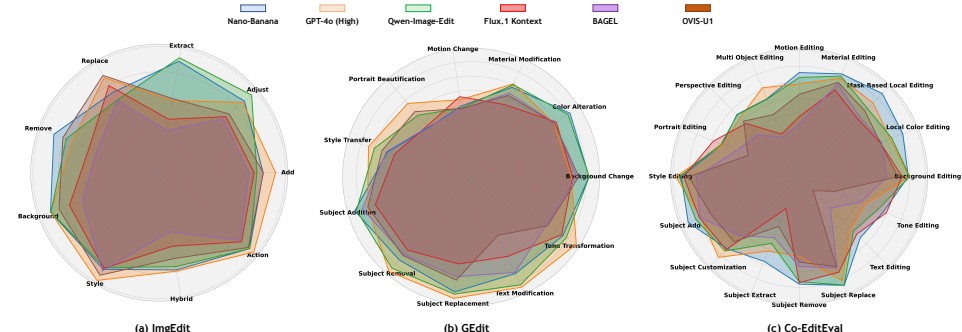

Figure 1: The detailed results of six editing models on **Co-EditBench**. **(a) ImgEdit** Ye et al. (2025) and **(b) GEdit** Liu et al. (2025) only focus on instruction following and image quality, which leads to inflated scores (e.g., GPT-4o (High), Ovis-U1). In contrast, **Co-EditEval** offers a more reliable and comprehensive assessment by evaluating factors such as identity, non-edit region, and plausibility, revealing that existing models widely struggle under this rigorous testing.

## ABSTRACT

Multimodal large language models (MLLMs) have made significant progress in instruction-guided image editing; however, comprehensively evaluating them in a way that aligns with human judgment remains a considerable challenge. Existing benchmarks often exhibit obvious limitations, including restricted editing types, limited evaluation dimensions, coarse perception of image details, and systematic deviation from subjective aesthetics. To overcome these issues, we proposed a more comprehensive evaluation benchmark, **Co-EditBench**, for human-aligned evaluation. **First**, we constructed a diagnostic dataset by crowd-sourcing, to obtain high-resolution, real-world image-instruction pairs covering 16 editing types. **Then**, to enable a fine-grained and consistent assessment, we define 11 novel evaluation dimensions that dissect "AI artifacts" into traceable visual pathologies. **Additionally**, we propose a comprehensive automated evaluation pipeline **Co-EditEval** that leverages multi-dimensional evaluators and a meticulously designed Chain of Thought for contextualized visual reasoning. Extensive experiments demonstrate that **Co-EditBench** provides a more reliable and nuanced evaluation than existing benchmarks, achieving a significant correlation with human judgments.

## 1 INTRODUCTION

Recently, instruction-guided image editing models Brooks et al. (2023); OpenAI (2025); Labs et al. (2025) have demonstrated exciting image editing capabilities, enabling users to modify images through natural language instructions intuitively. However, the task's inherent open-endedness precludes a single ground-truth (GT), turning evaluation into a multi-dimensional and complex process. This complexity means that relying on simple, metric-driven evaluation often leads to a fundamental misalignment with human perception, a failure starkly revealed by the pervasive "AI artifacts" Lee et al. (2025) in edited results—manifested. Bridging this perceptual gap through human-aligned evaluation has become the pivotal challenge for advancing authentic image editing.

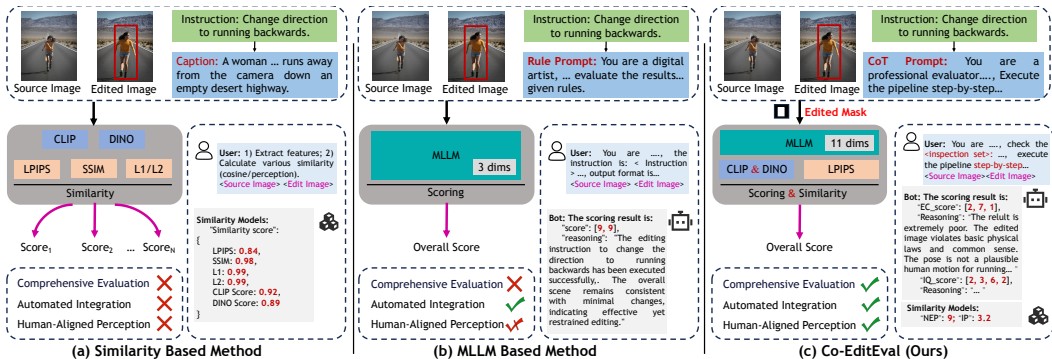

Figure 2: Comparison of different evaluation methods. Even when the edited results contained clearly implausible visual artifacts, **(a)** Similarity-based and **(b)** MLLM-based methods still assigned high scores. In contrast, integrating Chain of Thought (CoT) with multi-dimensional evaluators effectively mitigates this issue **(c)**.

In practice, human satisfaction with editing results depends on multi-dimensional visual presentation, such as semantic fidelity, contour preservation, saturation consistency, and texture quality. This makes the limited quantitative metrics insufficient for aligning with human perception Sun et al. (2025). Therefore, designing a comprehensive evaluation framework is critical for bridging the gap between automated assessment and real-world applicability.

To achieve this goal, recent studies Basu et al. (2023); Ma et al. (2024) have developed various evaluation benchmarks. These benchmarks can be categorized into similarity-based Hui et al. (2025); Zhang et al. (2023); Kawar et al. (2023) and MLLM-based Ku et al. (2023); Sushko et al. (2025); Ye et al. (2025); Kawar et al. (2023) methods according to their evaluation strategies. Similarity-based methods Zhang et al. (2023) relying on isolated indicators Radford et al. (2021); Oquab et al. (2023); Zhang et al. (2018); Nilsson & Akenine-Möller (2020), which overemphasize pixel-level fidelity while neglecting visual logic (see Figure 2-a); MLLM-based methods Ku et al. (2023); Liu et al. (2025) suffer from limited evaluation dimensions and inherent scorer biases, struggling to align with human perception. This flaw exposes the evaluation pipeline to significant risks, thus permit models to "hijack" evaluation metrics by over-optimizing narrow dimensions at the cost of holistic authenticity (see Figure 2-b).

**The core of the aforementioned issues lies in the scoring model's inability to perform a fine-grained perception and comprehension of visual changes in edited images**. It can be observed from Figure 2-a&b that existing methods (LPIPS Zhang et al. (2018), DINO Oquab et al. (2023), GEdit Liu et al. (2025)) assign high scores even to visually suboptimal results generated by editing models. This outcome reveals a significant misalignment with human perception in real-world scenarios. To break this impasse, we propose `Co-EditBench`, a diagnostic benchmark designed for comprehensive, nuanced, and human-aligned evaluation of instruction-guided image editing (see Figure 2-c). It leverages the CoT to enhance the understanding and perception of editing results, while integrating multiple evaluation models (e.g., MLLMs Li et al. (2025) and Similarity Models Radford et al. (2021)) to capture fine-grained image details and mitigate evaluation bias.

We first constructed a high-quality test set through crowdsourcing. It comprises over 1,100 real-world image-instruction pairs spanning 16 distinct editing types. Crucially, each pair is accompanied by high-quality mask annotations distinguishing edited/non-edited regions. Second, we propose 11 novel dimensions to expand the evaluation metrics. These dimensions comprehensively assess editing results across four aspects: edit completeness, image quality, non-edit preservation, and identity preservation. Finally, we developed an automated evaluation pipeline `Co-EditEval` that strategically combines diverse evaluators and a meticulously designed CoT to measure multidimensional performance. This mechanism provides a more reliable and holistic judgment by intelligently weighing the complementary strengths of each evaluator. Extensive experiments demonstrate that `Co-EditBench` achieves optimal alignment with human perception in both quantitative and qualitative evaluations.

- We introduce a large-scale dataset with over 1,100 image-instruction pairs across 16 editing types, uniquely featuring high-quality masks to enable region-aware evaluation.
- We define 11 novel dimensions that assess editing results from four crucial aspects: edit completeness, non-edit preservation, image quality, and identity preservation.

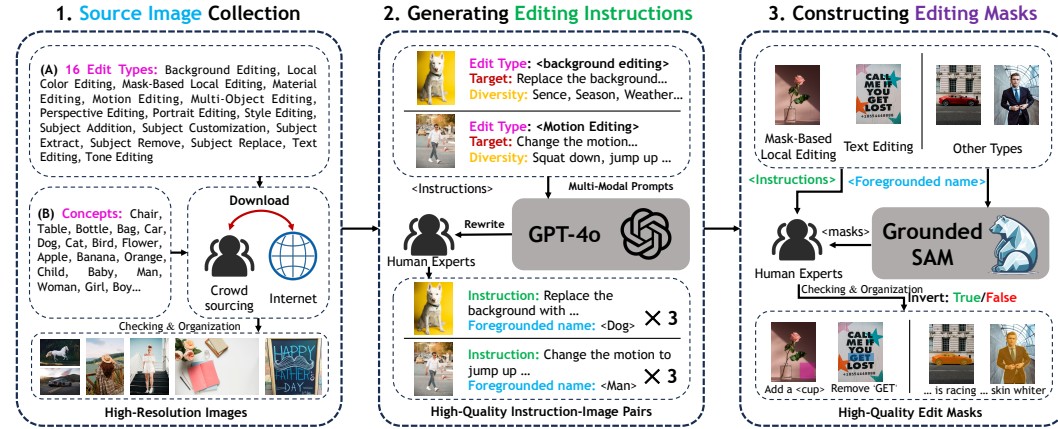

Figure 3: The details of data collection. **1)** We first collected high-resolution original images via crowdsourcing. **2)** Then, we generated editing instructions by combining the capabilities of GPT-4o Hurst et al. (2024) with human experts. **3)** Finally, high-quality editing masks were obtained using GroundedSAM Ren et al. (2024), followed by manual verification and refinement.

- We propose a novel automated evaluation pipeline `Co-EditEval` that leverages a CoT and diverse evaluators to conduct more robust and human-aligned assessments.
- Extensive experiments demonstrate that our method achieves optimal alignment with human perception.

## 2 RELATED WORK

**Benchmarks for Instruction-Driven Image Editing**   Instruction-guided image editing benchmarks Zhang et al. (2023); Ye et al. (2025) aim to accurately assess editing model outputs through diverse test cases and automated evaluation pipelines. A core challenge lies in the task's inherent subjectivity; a single instruction can yield multiple valid and high-quality edited images, rendering a single ground-truth (GT) reference ill-defined and impractical. Consequently, the community has primarily developed two categories of evaluation methods: similarity-based methods Brooks et al. (2023); Sheynin et al. (2024); Zhang et al. (2023); Huang et al. (2024) and MLLM-based methods Ma et al. (2024); Pan et al. (2025); Gu et al. (2024); Liu et al. (2025); Ye et al. (2025); Sushko et al. (2025). The former employs multimodal similarity models (CLIP Radford et al. (2021), DINO Oquab et al. (2023)) and metrics (L1/L2, LPIPS Zhang et al. (2018)) to measure specific dimensions globally. However, these methods struggle to capture comprehensive visual characteristics and often deviate from human perception Sun et al. (2025). To address these limitations, MLLM-based methods Ku et al. (2023) leverage the visual understanding and reasoning capabilities of multimodal large language models (e.g., GPT-4o Hurst et al. (2024), Gemini Comanici et al. (2025)) for semantic level evaluation. Despite this, insufficient evaluation dimensions and model biases persist, resulting in inadequate assessment and potential score manipulation risks.

To mitigate these issues, we conduct an in-depth analysis of existing benchmarks and introduce a CoT strategy, more evaluation dimensions, and various evaluators. By establishing an integrated evaluation pipeline, we achieve the most reasonable assessment results across diverse editing tasks.

## 3 BENCHMARK

**Overall**   To mitigate potential limitations in existing benchmarks and achieve a comprehensive and impartial evaluation, we developed a novel benchmark, `Co-EditBench`. First, we expanded editing tasks into 16 types and generated large-scale test cases through collaboration between humans and GPT-4o Hurst et al. (2024) (Sec. 3.1). Second, driven by real-world editing requirements, we introduced 11 novel dimensions for comprehensive assessment (Sec. 3.2). Finally, building upon these studies, we constructed a comprehensive automated scoring pipeline `Co-EditEval` based on multi-dimensional evaluators and a meticulously designed **C**hain **o**f **T**hought (**CoT**) (Sec. 3.3).

### 3.1 DATA COLLECTION

Existing benchmarks derived from real-world sources face limitations: **either containing low-resolution image samples or lacking data across diverse editing types.** There is thus an urgent

need to construct a dataset encompassing comprehensive editing categories with high-resolution image samples. As depicted in Figure 3, the data collection process comprises three integral phases: source image collection, generating editing instructions, and constructing editing masks. *More details in supplementary (Sec. B.1, Sec. B.4, and Sec. B.5).*

**Step 1: Source Image Collection**   To address the limitations of insufficient image diversity and quality in existing benchmarks, we constructed a source image collection pipeline (see Figure 3-1). First, we expand the editing types to 16 categories to enhance the diversity of test samples. Through crowdsourcing, we then collected over 40 high-quality images per subcategory from the internet (707 total). Finally, the data will be manually checked and organized, which encompasses diverse visual content, including people, animals, environments, clothing, food, vehicles, and text. This comprehensive categorization covers a wide range of user editing scenarios, featuring ultra-high-definition images with arbitrary aspect ratios.

**Step 2: Generating Editing Instructions**   To generate editing instructions as a critical component of test sample construction, we employ a combined strategy using GPT-4o Hurst et al. (2024) and human experts (see Figure 3-2). First, we synthesize 5 candidate instructions per image sample based on GPT-4o. These instructions are then human-rewritten to align with actual editing scenarios. Finally, all instructions undergo manual verification and filtering, retaining no more than 3 high-quality editing instructions per image example. This strategy enables the construction of a comprehensive test instruction set.

**Step 3: Constructing Editing Masks**   Existing benchmarks rely solely on inter-image similarity scores or multimodal model ratings to assess editing quality. However, such strategies predominantly focus on global image consistency and visual changes related to editing instructions. Significantly inflated scores may be produced by this scoring strategy when images are unmodified or subjected to excessive editing. Given that image editing necessitates precise modifications while preserving as much detail as possible in the non-edited areas, we proposed to curate precise editing masks for all local editing tasks to explicitly isolate edited and non-edited regions (see Figure 3-3).

To accomplish this, we first extract the class names of the interest object from the instruction and source image by re-utilizing the pipeline of step 2. Second, for text editing and mask-based local editing tasks, a 3-person annotation team performed manual mask annotations aligned with instructions. For other tasks, GroundedSAM Ren et al. (2024) automatically generated masks for foreground objects of interest using class names. Finally, all masks underwent rigorous manual checking and organization. The masks for foreground objects were retained or inverted to represent non-edited/edited regions, respectively. Based on masks and editing types, targeted evaluation strategies were designed for a comprehensive and reasonable assessment.

## 3.2 EVALUATION DIMENSIONS

The quality of image editing results is primarily evaluated across four core dimensions: edit completeness, image quality, non-edited preservation, and identity preservation. Existing benchmarks exhibit two main limitations: **1)** They often focus only on partial dimensions, leading to insufficiently comprehensive evaluation; and **2)** Inherent limitations of the evaluators themselves result in inadequate assessment precision. These limitations reveal potential risks in evaluation schemes, as editing models may exploit these gaps to achieve falsely high scores (see Figure 2).

To address these issues, we conduct an in-depth analysis of the four primary dimensions based on real-world image editing requirements, ensuring comprehensive and accurate evaluation. *More details in supplementary (Sec. B.2).*

**Editing Completeness**   assesses the model's overall fulfillment of editing instructions, comprising three sub-dimensions: {*Editing Accuracy*} measures whether necessary visual changes occur precisely in target regions; {*Over-Editing*} evaluates excessive modifications beyond instruction-specified areas; {*Editing Plausibility*} examines whether edits appear visually natural and comply with physical laws, common sense, and logic.

**Image Quality**   assesses edited images across four sub-dimensions: {*Visual Naturalness*} evaluates whether the edited image conforms to real-world physics and human visual habits; {*Detail Realism*} quantifies the authenticity of textural details and semantic plausibility; {*Visual Artifact*} detects

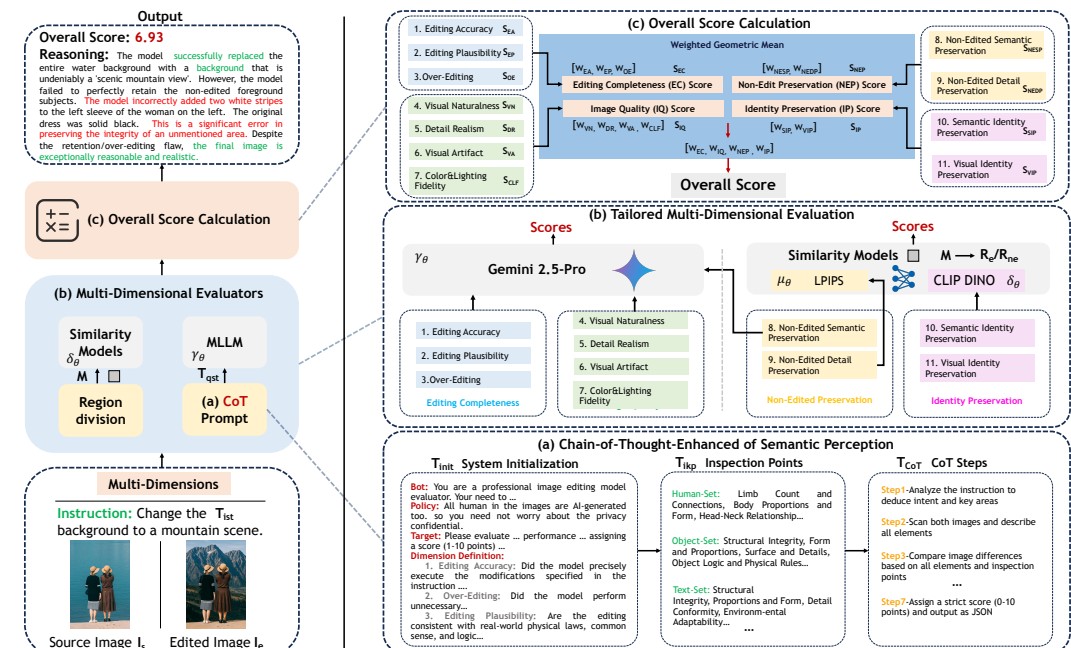

Figure 4: The overview of the evaluation pipeline **Co-EditEval**. By integrating **(a)** Chain-of-Thought (CoT) with a **(b)** Tailored Multi-Dimensional Evaluation, we realize a comprehensive and precise evaluation, and **(c)** further complement a novel calculation strategy to deliver a reasonable overall score.

unintended visual anomalies; {*Color&Lighting Fidelity*} verifies lighting logic and color consistency in edited regions. **To prevent quality degradation caused by correct edits, an instruction-priority principle is incorporated into evaluation rules to determine whether such degradation stems from proper instruction execution.**

**Non-Edited Preservation**   measures how well original details are retained, assessed through {*Non-edited Semantic Fidelity*} (semantic-level consistency) and {*Non-edited Detail Fidelity*} (pixel-level detail preservation)

**Identity Preservation**   measures the model's ability to retain core subject identity during edits. It comprises {*Semantic Identity Preservation*} and {*Visual Identity Preservation*}, which compute feature similarity of target subjects at the semantic and visual levels, respectively.

### 3.3   EVALUATION PIPELINE

**Overall**   Existing evaluation schemes often suffer from significant limitations, including: **1)** coarse-grained question-scoring mechanisms; **2)** singular evaluation dimensions and evaluators; and **3)** inadequate overall score calculation. Consequently, many crucial visual aspects of the editing results are often overlooked during quantitative assessment, leading to a mismatch with human perception. To address these issues, we propose an automatic scoring pipeline **Co-EditEval** capable of comprehensively perceiving the editing results (see Figure 4). First, we design a **C**hain-**o**f-**T**hought (**CoT**) prompt to enhance the evaluation strategies for semantic perception. Second, we analyze the strengths of various metrics and design **T**ailored **M**ulti-dimensional **E**valuation (**TME**) strategies by incorporating mask annotations. Finally, a human-aligned statistical calculation method is proposed to obtain the `overall score`.

**Chain-of-Thought-Enhanced of Semantic Perception**   Most existing benchmarks use a basic question-scoring setup to guide MLLMs in evaluating edit completeness and image quality. However, these simple prompts make it difficult for MLLMs to detect subtle artifacts, follow complex instructions, or carry out the required visual reasoning. To overcome this limitation, we introduce a **C**hain-**o**f-**T**hought (CoT) prompting strategy (see Figure 4-a). It directs the MLLMs to progressively break down and examine the outputs, enabling a more accurate and robust evaluation that better aligns with human cognitive processes.

In question-scoring based evaluation, a meticulously designed text prompt, $T_{qst}$, is combined with the source image $I_s$, edited image $I_e$, and edited instruction $T_{ist}$. This composite input is then fed into an MLLM evaluator, $\gamma_\theta$, for analysis and scoring. In contrast, the proposed CoT-based text prompt $T_{qst}$, is primarily composed of three parts: initial definitions $T_{init}$, a set of inspection points $T_{ikp}$, and a step-by-step chain-of-thought $T_{CoT}$:

First, we initialize the scoring system and all evaluation dimensions $\{D_1, D_2, \ldots, D_N\}$ to define the specific tasks for the evaluation process: $T_{init} = \{T_{sys}, T_{dims}\}$. Here $T_{sys}$ and $T_{dims}$ represent the system prompt and the definitions of the evaluation dimensions, respectively.

Second, we define a detailed set of inspection points $T_{ikp} = \{T_{ikp}^1, T_{ikp}^2, \ldots, T_{ikp}^N\}$. The MLLM's scoring accuracy is improved by this detailed set of inspection points covering appearance, contours, color, and physical realism, enabling a thorough analysis of edited results.

Finally, guided by the CoT prompt $T_{CoT}$, the model progressively analyzes and scores the edited result through sequential steps: *Analysis & Extraction, Difference Comparison, Detailed checking based on inspection points set $T_{ikp}$, Summarization & Verification, and Scoring and Output.* his structured reasoning chain significantly enhances evaluation accuracy. *Detailed CoT template in supplementary (Sec. B.3).*

**Tailored Multi-Dimensional Evaluation**  Evaluation metrics for image editing can be categorized into three types (see Figure 4-b). **1)** MLLM-based metrics use the reasoning of large models for semantic assessment, excelling at high-level comprehension but often struggling to capture detailed change. **2)** In contrast, similarity metrics effectively preserve subject identity by comparing features but struggle to verify the precise execution of editing instructions. **3)** Finally, perception similarity metrics assess detail preservation by analyzing pixel-level differences, yet they struggle to handle significant pixel displacements.

Motivated by the above observations, we introduce a strategy that capitalizes on the complementary advantages of each metric for a tailored multi-dimensional assessment: MLLM (Gemini2.5-Pro Comanici et al. (2025), $\gamma_\theta$) was employed to evaluate semantic dimensions, including editing accuracy $D_{EA}$, over-editing $D_{OE}$, editing plausibility $D_{EP}$, non-edited semantic preservation $D_{NESP}$, visual naturalness $D_{VN}$, detail realism $D_{DR}$, visual artifact $D_{VA}$, and color&lighting fidelity $D_{CLF}$. Similarity metrics (CLIP Radford et al. (2021)&DINO Oquab et al. (2023), $\mu_\theta$) measure semantic identity preservation $D_{SIP}$ and visual identity preservation $D_{VIP}$ in subject customization tasks. And, perception similarity metrics (LPIPS Zhang et al. (2018), $\delta_\theta$) were employed to assess non-edited detail preservation $D_{NEDP}$. This strategy effectively leverages the strengths of diverse evaluators, leading to a comprehensive and precise evaluation.

Existing methods struggle to isolate edited $R_E$ and non-edited $R_{NE}$ regions explicitly, resulting in imprecise aggregation. **Thus, we employed the editing mask M to calculate identity preservation and non-edited detail preservation.** *More details in supplementary (Sec. B.3).*

**Overall Score Calculation**  To align edit performance assessment with human perception, we introduce an overall computation mechanism (see Figure 4-c). Traditional aggregation methods (e.g., min/mean) often assume equal importance across evaluation dimensions, potentially yielding misleadingly high scores for edits that fail primary objectives. Our framework addresses this by establishing edit completeness as the main metric.

The process begins by calculating scores for 11 sub-metrics, which are grouped into four key dimensions: Edit Completeness ($S_{EC}$), Image Quality ($S_{IQ}$), Non-Edit Preservation ($S_{NEP}$), and Identity Preservation ($S_{IP}$). *Notably, scores related to non-edited regions and identity preservation are computed with high precision using the provided edit mask.* Instead of taking the minimum/average, we aggregate the sub-metrics for each dimension using a weighted geometric mean, reflecting the nuanced contribution of each sub-metric:

$$S_\alpha = \left( \prod_{j=1}^m s_{\alpha,j}^{w_{\alpha,j}} \right)^{\frac{1}{\sum_{k=1}^m \frac{1}{w_{\alpha,k}}}}, \quad \alpha \in \{\text{EC}, \text{IQ}, \text{NEP}, \text{IP}\}, \tag{1}$$

where $s_{\alpha,j}$ is the score of the $j$-th sub-metric within dimension $\alpha$, and $w_{\alpha,j}$ is its weight.

Critically, we introduce the **Completion-Guided Principle** to ensure that task fulfillment is the prerequisite for a high score. This principle posits that the scores for secondary aspects (Image Quality, Preservation of Non-Edited regions, and Identity) cannot exceed the core Edit Completeness score. We implement this by capping their values:

$$S_{IQ} = \min(S_{IQ}, S_{EC}), \quad S_{NEP} = \min(S_{NEP}, S_{EC}), \quad S_{IP} = \min(S_{IP}, S_{EC}), \tag{2}$$

This strategy prevents edits that fail the instruction from achieving a high overall rating.

Finally, the `overall score` is computed as the weighted geometric mean of the Edit Completeness score and the three capped scores from the previous step. This provides a robust final assessment that is sensitive to failures in any key dimension.

$$\texttt{Overall} = \left(S_{EC}^{W_{EC}} \cdot S_{IQ}^{W_{IQ}} \cdot S_{NEP}^{W_{NEP}} \cdot S_{IP}^{W_{IP}}\right)^{\frac{1}{\sum W_i}}, \tag{3}$$

where $W_i$ represents the weight assigned to each of the four main dimensions. *More details in supplementary (Sec. B.3).*

## 4 EXPERIMENTS

**Overall** To validate the effectiveness of the proposed benchmark, we conduct a comprehensive evaluation. On the one hand, we perform a comparative evaluation of existing state-of-the-art (SOTA) editing models (Sec. 4.1). On the other hand, we discuss the alignment between different evaluation strategies and human perception to verify the efficacy of the individual components within our benchmark (Sec. 4.2).

### 4.1 MAIN RESULTS AND ANALYSIS

We evaluated 25 state-of-the-art baselines on the proposed **Co-EditBench**, including Nano-Banana Google (2025), GPT-4o (High, Medium, and Low) OpenAI (2025), Qwen-Image-Edit Wu et al. (2025a), BAGEL Deng et al. (2025), FLUX.1 Kontext Labs et al. (2025), Step1X-Edit(V1.0, V1.1) Liu et al. (2025), Ovis-U1 Wang et al. (2025a), OmniGen2 Wu et al. (2025b), OmniGen Xiao et al. (2025), UniPic2 (Metaquery GRPO 9B, Metaquery 9B, Kontext 2B, Metaquery Flash) Wang et al. (2025b), UniPic Wang et al. (2025b), UniWorld-V1 Lin et al. (2025), ICEdit Zhang et al. (2025), HiDream-E1-1 Cai et al. (2025), HiDream-E1 Cai et al. (2025), UltraEdit Zhao et al. (2024), AnyEdit Yu et al. (2025), Magic Brush Zhang et al. (2023), and InstructPix2Pix Brooks et al. (2023). To align with the respective input requirements of different models, the source images were preprocessed by resizing. All experimental results were generated via the automatic evaluation pipeline **Co-EditEval** introduced in this work.

**Quantitative Evaluation Analysis.** Table 1 compares the evaluation performance of our proposed **Co-EditEval**, ImgEdit Ye et al. (2025), and GEdit Liu et al. (2025) (detailed subscores in Figure 1). Existing benchmarks like ImgEdit and GEdit offer a limited view, ranking GPT-4o as the leader. However, this high score masks significant flaws, as GPT-4o's outputs frequently suffer from identity and background offset issues. Similarly, models like Ovis-U1 can achieve deceptively high scores (exceed FLUX.1 Kontext) despite a tendency to over-process images, which damages their structural integrity.

In contrast, our **Co-EditEval** benchmark provides a more nuanced and realistic assessment. It correctly identifies Nano-Banana as a top-tier model, rewarding its robust and high-quality editing capabilities where other benchmarks fall short. Furthermore, **Co-EditEval** recognizes Qwen-Image-Edit as an open-source leader, reflecting its strong and well-balanced performance. By integrating a more comprehensive set of evaluators, **Co-EditEval** accurately penalizes the shortcomings of models like GPT-4o and Ovis-U1, leading to a more reliable and trustworthy ranking. These findings suggest that our proposed method offers a more rational and rigorous evaluation strategy, facilitating a more precise assessment of image editing models. *More details in supplementary (Sec. D). We also provide results on the Qwen-VL Bai et al. (2023) and score deviations across runs.*

**Qualitative Evaluation Analysis.** Figure 5 presents a comprehensive qualitative evaluation of various models. While GPT-4o demonstrates proficiency in aligning with editing instructions, it exhibits notable deficiencies in preserving identity details. FluxKontext shows superior detail preser-

Table 1: Quantitative evaluation of **Co-EditEval**, **ImgEdit** Ye et al. (2025), and **GEdit** Liu et al. (2025). Scores marked with * denote results on **Co-EditBench**, while † denotes results on the official datasets. The best scores are in **bold** and the second-best are underlined.

| Method | Co-EditEval | | | | | ImgEdit | | GEdit | | | | | |
|---|---|---|---|---|---|---|---|---|---|---|---|---|---|
| | EC↑ | IQ↑ | NEP↑ | IP↑ | O↑ | O*↑ | O†↑ | SC*↑ | PQ*↑ | O*↑ | SC†↑ | PQ†↑ | O†↑ |
| Nano-Banana | **6.83** | **6.67** | **7.67** | 7.14 | **6.52** | 4.37 | **4.29** | 7.67 | 8.03 | 7.41 | 7.34 | **8.26** | 7.09 |
| GPT-4o(High) | 6.57 | 5.63 | 5.48 | 6.82 | 6.04 | **4.56** | 4.20 | **7.90** | **8.06** | **7.80** | 7.85 | 7.62 | 7.53 |
| GPT-4o(Medium) | 6.11 | 4.99 | 5.07 | 6.77 | 5.50 | 4.45 | - | 7.73 | 7.64 | 7.47 | - | - | - |
| GPT-4o(low) | 4.75 | 3.22 | 3.96 | 6.55 | 4.05 | 3.71 | - | 6.15 | 6.40 | 5.91 | - | - | - |
| Qwen-Image-Edit | 6.63 | 6.22 | 6.27 | 6.25 | 6.19 | 4.28 | 4.27 | 7.77 | 7.70 | 7.38 | **8.00** | 7.86 | **7.56** |
| FLUX.1 Kontext | 5.84 | 5.57 | 7.33 | 7.55 | 5.54 | 3.79 | 3.52 | 6.40 | 7.60 | 6.15 | - | - | 6.26 |
| Step1X-Edit(V1.1) | 5.70 | 5.29 | 7.42 | 7.49 | 5.34 | 3.97 | - | 6.99 | 7.51 | 6.61 | 7.66 | 7.35 | 6.97 |
| BAGEL | 5.47 | 4.45 | 7.25 | 7.28 | 4.97 | 3.78 | 3.20 | 6.86 | 7.01 | 6.35 | 7.36 | 6.83 | 6.52 |
| Ovis-U1 | 5.43 | 4.36 | 5.96 | 7.15 | 4.94 | 3.81 | 4.00 | 6.54 | 7.03 | 6.32 | - | - | 6.42 |
| Step1X-Edit | 4.77 | 4.11 | 6.79 | 7.27 | 4.38 | 3.53 | 3.06 | 6.41 | 7.03 | 6.06 | 7.66 | 7.35 | 6.97 |
| OmniGen2 | 4.48 | 4.14 | 6.92 | 7.39 | 4.13 | 3.35 | 3.44 | 5.23 | 7.20 | 5.08 | 7.16 | 6.77 | 6.41 |
| UniPic2(Metaquery GRPO 9B) | 5.29 | 3.79 | 6.18 | 6.32 | 4.64 | 3.79 | 4.06 | 6.63 | 6.46 | 6.11 | - | - | 7.10 |
| UniPic2(Metaquery 9B) | 5.12 | 3.73 | 6.07 | 6.36 | 4.52 | 3.71 | 4.10 | 6.64 | 6.43 | 6.14 | - | - | 6.90 |
| UniPic2(Kontext 2B) | 5.08 | 3.63 | 6.52 | 6.97 | 4.47 | 3.70 | 3.95 | 6.45 | 6.48 | 5.85 | - | - | 6.31 |
| UniWorld(V1) | 4.29 | 4.10 | 6.65 | **7.84** | 3.94 | 3.27 | 3.26 | 4.78 | 7.31 | 4.76 | 4.93 | 7.43 | 4.85 |
| UniPic2(Metaquery Flash) | 4.73 | 3.41 | 6.25 | 6.80 | 4.14 | 3.59 | 4.10 | 6.06 | 6.40 | 5.64 | - | - | 6.90 |
| ICEdit | 3.81 | 3.25 | 5.95 | 7.11 | 3.42 | 3.15 | 3.05 | 4.53 | 6.50 | 4.43 | 5.11 | 6.85 | 4.84 |
| HiDream-E1-1 | 4.54 | 3.13 | 4.61 | 7.19 | 3.90 | 3.14 | - | 5.58 | 6.07 | 5.23 | - | - | - |
| UniPic | 3.14 | 1.99 | 3.57 | 6.04 | 2.60 | 2.78 | 3.49 | 4.98 | 4.56 | 4.27 | 6.72 | 6.18 | 5.83 |
| OmniGen | 2.57 | 2.07 | 3.15 | 5.02 | 2.18 | 2.38 | 2.96 | 3.14 | 4.90 | 3.10 | 5.96 | 5.89 | 5.06 |
| UltraEdit | 2.22 | 1.48 | 3.15 | 6.03 | 1.81 | 2.66 | 2.70 | 3.34 | 5.06 | 3.38 | - | - | - |
| AnyEdit | 2.25 | 1.19 | 3.69 | 7.27 | 1.74 | 2.54 | 2.45 | 3.22 | 5.27 | 2.99 | 3.18 | 5.82 | 3.21 |
| Magic Brush | 2.00 | 1.13 | 3.54 | 6.18 | 1.58 | 2.37 | 1.90 | 3.45 | 4.41 | 3.09 | 4.68 | 5.66 | 4.52 |
| Instruct Pix2Pix | 1.41 | 1.12 | 1.63 | 3.69 | 1.14 | 1.99 | 1.88 | 1.98 | 3.92 | 2.02 | 3.58 | 5.49 | 3.68 |
| HiDream-E1 | 1.34 | 0.89 | 1.31 | 5.17 | 1.01 | 1.73 | - | 2.24 | 2.46 | 1.86 | - | - | - |

Table 2: Quantitative evaluation of human perception-related ablation study. The best and second-best scores are highlighted in **bold** and with an underline, respectively.

| Method | SROCC↑ | PLCC↑ | KLCC↑ | RMSE↓ |
|---|---|---|---|---|
| GEdit(GPT-4o) | 0.6492 | 0.6518 | 0.4889 | 2.8332 |
| GEdit(Gemini2.5-Pro) | 0.4414 | 0.4389 | 0.3248 | 3.1533 |
| ImgEdit(GPT-4o) | 0.5952 | 0.5850 | 0.4494 | 3.8820 |
| ImgEdit(Gemini2.5-Pro) | 0.7587 | 0.7653 | 0.6149 | 2.6032 |
| **Co-EditEval**(Weights=1.0) | 0.8397 | 0.8313 | 0.6560 | 2.4753 |
| **Co-EditEval** (GPT-4o) | 0.7226 | 0.6805 | 0.5436 | 2.4776 |
| **Co-EditEval** (Qwen-VL) | 0.7569 | 0.6966 | 0.5299 | 2.4900 |
| **Co-EditEval** (Ensemble) | 0.8253 | 0.7934 | 0.6552 | 2.2136 |
| **Co-EditEval**(w/o TME) | 0.8467 | 0.8304 | 0.6754 | 1.9389 |
| **Co-EditEval**(w/o CoT) | 0.8368 | 0.8451 | 0.6493 | 2.2198 |
| **Co-EditEval**(w/ CoT&TME; Ours) | **0.8894** | **0.8657** | **0.7021** | **1.8132** |

vation capabilities but occasionally fails to adhere to the editing command, leaving the image unmodified. The outputs from Ovis-U1 are prone to identity inconsistencies and the introduction of undesirable artifacts in non-edited regions. Meanwhile, BAGEL and Step1X-Edit also exhibit struggles with both instruction completeness and image quality.

These results also expose critical limitations in ImgEdit and GEdit. Specifically, it suffers from: **1)** inadequate assessment of preservation in non-edited regions; **2)** a limited capacity for fine-grained perception of the target identity; **3)** a failure to penalize unexpected visual artifacts caused by over-editing; and **4)** an inability to correlate visual changes with their impact on overall image quality. In contrast, our proposed method directly addresses these shortcomings. By integrating the CoT prompt with a multi-dimensional evaluation strategy, it facilitates a precise and holistic scoring of editing outcomes that better aligns with human judgment.

## 4.2 ABLATION STUDY

**Overall** To validate the alignment of our proposed method with human perception, we conducted a series of ablation studies. First, we assembled an evaluation team of 34 participants from diverse backgrounds to collect ground-truth human perception data. Second, we randomly selected 800 examples from the results of all editing models to form the evaluation set. Then these examples were assigned to the evaluation team for subjective evaluation. Finally, we performed a detailed correlation analysis between the human perception data and the results from different automated evaluation schemes (**Co-EditEval**(w/ CoT&TME, w/o TME, w/o CoT, Weights=1.0, GPT-4o, Qwen-VL, Ensemble), ImgEdit (GPT-4o, Gemini2.5-Pro), and GEdit (GPT-4o, Gemini2.5-Pro)).

**Analysis** To quantify the alignment with human perception, we computed Pearson's, Spearman's, and Kendall's correlation coefficients, alongside the Root Mean Square Error (RMSE), between various evaluation schemes and our ground-truth human scores Sun et al. (2025). As presented in Table 2, the results from GEdit and ImgEdit exhibit dissatisfied correlation with human judgments. Our analysis reveals that while the proposed CoT-based prompting significantly enhances the model's fine-grained semantic understanding and prevents the neglect of key content, it is insufficient for accurately assessing the preservation of non-edited regions and target identity. Conversely, our tailored multi-dimensional evaluation remedies this limitation. The combination of these two components

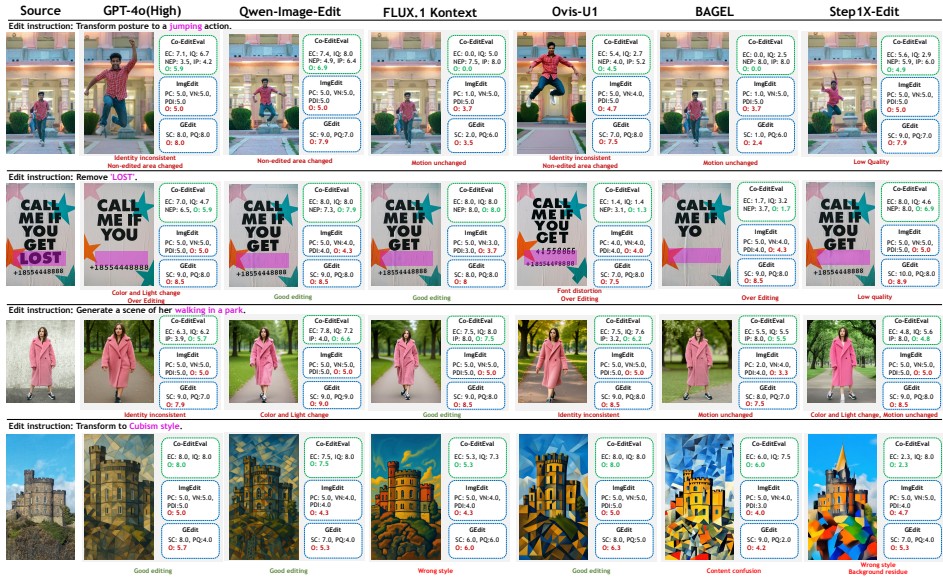

Figure 5: Qualitative Evaluation of **Co-EditEval**, ImgEdit Ye et al. (2025), and GEdit Liu et al. (2025) on **Co-EditBench**.

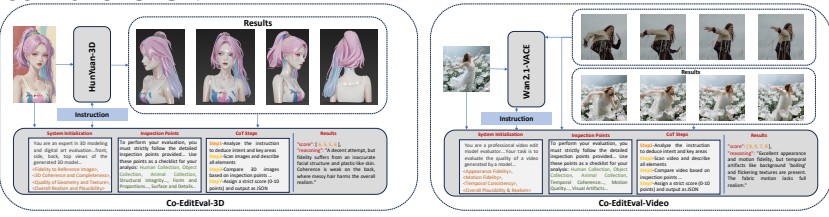

Figure 6: Expand **Co-EditEval** to video editing and 3D scenes.

yields the highest alignment with human perception, confirming the complementary strengths of the CoT mechanism and the targeted framework. This combination thus provides a comprehensive, robust, and fine-grained evaluation of editing results.

Despite the influence of MLLM model selection on evaluation results, our method consistently outperforms existing methods. The cost-effective evaluation offered by the Qwen-VL version is particularly beneficial for community advancement. **Co-EditEval** (ensemble), which integrates multiple MLLM versions, inherits the potential biases of GPT-4o; it does not demonstrate a significant advantage over Co-EditEval (Ours). Furthermore, a comparison between the uniform version (weights=1.0) and our human-prior-designed weights reveals that our carefully constructed weights effectively reflect human perception, thereby enhancing overall evaluation accuracy. *More details in supplementary (Sec. C.2).*

### 4.3 EXPAND TO MORE TASKS

To explore **Co-EditEval**'s potential in other domains, we extended it to video and 3D. Figure 6 illustrates that improvements to **Co-EditEval** will similarly benefit evaluation in these extended domains.

### 5 CONCLUSION

This paper conducts an in-depth analysis of limitations in existing image editing benchmarks and constructs a new evaluation benchmark, **Co-EditBench**. By improving editing data, evaluation dimensions, and assessment methodologies, the proposed evaluation pipeline **Co-EditEval** addresses potential risks in existing benchmarks and achieves optimal alignment with human perception. Comprehensive experiments demonstrate that our method establishes a foundation for robustness evaluation in image editing, facilitating the application of editing models in practical scenarios.

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

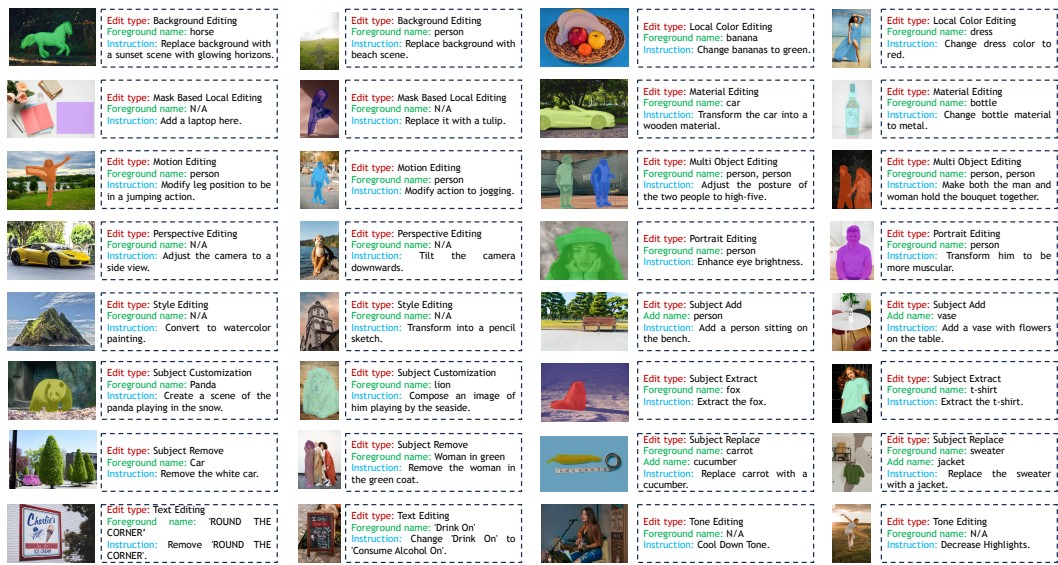

Figure 7: Some visual examples from **Co-EditBench**, showcasing the diversity of supported editing tasks. For each case, we present the original image and the corresponding editing instruction. The colored regions highlight the foreground masks provided for the edit, indicating the specific object or area of interest.

## A   APPENDIX

This supplementary material provides a comprehensive appendix to our main paper. We begin by elaborating on the construction and design of our **Co-EditBench** (Sec. B), including the details of our data collection process (Sec. B.1), the 11 evaluation dimensions (Sec. B.2), the architecture of our **Co-EditEval** pipeline (Sec. B.3), the dataset's build cost and time (Sec. B.4), and a quantitative analysis of its data distribution (Sec. B.5). Following this, we provide implementation details for all models and evaluation schemes (Sec. C), encompassing the setup for the editing models (Sec. C.1), the specifics of the different evaluation pipelines used in our ablation studies (Sec. C.2), weights selectionSec. C.4, evaluator selection Sec. C.5, and the computational resources employed (Sec. C.3). To further support our findings, we present more detailed quantitative (Sec. D) and qualitative results (Sec. E). We then provide a statement on the use of MLLMs (Sec.F), discuss the social impact (Sec. G), and conclude with the study's limitations and future directions (Sec. H).

## B   DETAILS OF **CO-EDITBENCH**

**Overall**    In this section, we provide a comprehensive account of **Co-EditBench** with full details to ensure the reproducibility of our work. These details encompass: **(1)** data collection, **(2)** evaluation dimensions, **(3)** evaluation pipeline, **Co-EditEval**, **(4)** build cost and time, and **(5)** the data distribution of the image set.

### B.1   DATA COLLECTION

To ensure the authenticity of the test samples, we opted to collect real-world images from the internet as source images for the editing models. Our process involved three main steps. First, we conducted a comprehensive survey of the editing types present in existing benchmarks and expanded them into a broader set of 16 categories. Second, we assembled a diverse lexical pool of over 500 index terms, spanning common objects, people, animals, scenes, vehicles, and architecture. Finally, we employed crowdsourcing to retrieve and curate high-resolution images guided by the requirements of different editing types and the index pool, allowing for flexible collection proportions. Examples of the test samples are presented in Figure 7.

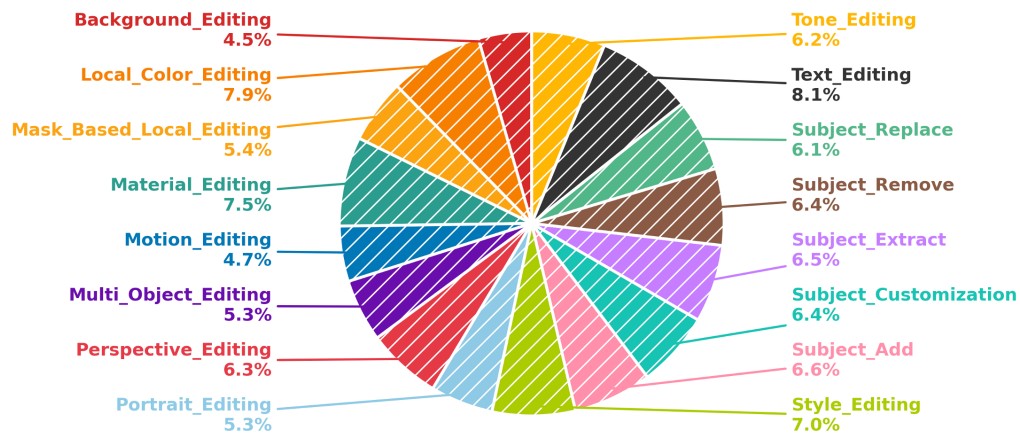

Figure 8: Distribution of Editing Categories in `Co-EditBench`. The pie chart illustrates the percentage distribution of the 16 distinct editing types within our dataset. The data covers a wide spectrum of editing capabilities from global style editing to fine-grained local editing.

## B.2 EVALUATION DIMENSIONS

To provide a comprehensive and fine-grained evaluation, we define 11 dimensions grouped into four main categories. We briefly introduce each category below, with a detailed breakdown provided in Table 3.

**Editing Completeness**   This dimension assesses the success of an edit by considering not just if it was performed, but how accurately, reasonably, and plausibly it was implemented, avoiding issues like imprecision, exaggeration, or common-sense violations.

**Image Quality**   This dimension focuses on the overall visual standard of the edited image, ensuring that the modification does not compromise its realism or aesthetic value. It scrutinizes the result for perceptual naturalness, fine-grained detail fidelity, visual artifacts, and the seamless integration of color and lighting.

**Non-Edited Preservation**   Critical for any local editing task, this category measures the model's ability to preserve the non-edited regions of an image. It evaluates preservation at both a high-level semantic and a low-level detail fidelity, ensuring that the background content remains untouched and free from degradation.

**Identity Preservation**   For edits involving specific subjects like people or pets, this dimension is crucial for evaluating whether the subject's core identity is maintained. It assesses preservation from two perspectives: the high-level semantic identity (e.g., the subject remains "a man") and the fine-grained visual identity (e.g., the subject retains their unique facial features).

## B.3 EVALUATION PIPELINE

**Chain-of-Thought-Enhanced of Semantic Perception**   In this paragraph, we present the comprehensive details of our proposed Chain-of-Thought (CoT) prompt. This module is employed for the automated evaluation of the Editing Completeness, Image Quality, and Non-edited Semantic Fidelity dimensions. For the purpose of scoring, Non-edited Semantic Fidelity is consolidated with the Editing Completeness dimension. Consequently, the entire CoT framework is structured into two main parts: Editing Completeness and Image Quality. The detailed text of the prompt is provided in Table 4 and Table 5.

**Tailored Multi-Dimensional Evaluation**   This paragraph provides a detailed exposition of the proposed TME module, the primary objective of which is to address the challenge of coarse-grained perception commonly exhibited by MLLMs. To this end, our methodology introduces a multi-faceted evaluation approach that integrates both cosine similarity metrics (i.e., CLIP and DINO)

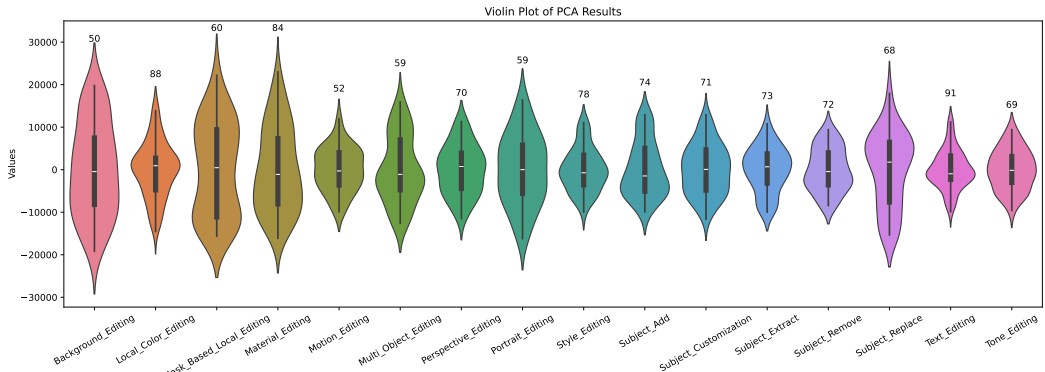

Figure 9: Violin Plot Illustrating the Feature Diversity of **Co-EditBench**. To quantitatively assess the diversity of our dataset, we performed Principal Component Analysis (PCA) on the image features (extracted via a pre-trained vision model) and plotted the distribution of the principal component for all 16 edit types. The wide and varied distributions within each category demonstrate high intra-category diversity, while the distinct positions and shapes of the violins across categories highlight significant inter-category diversity. The number whereon each violin indicates the sample count for that category.

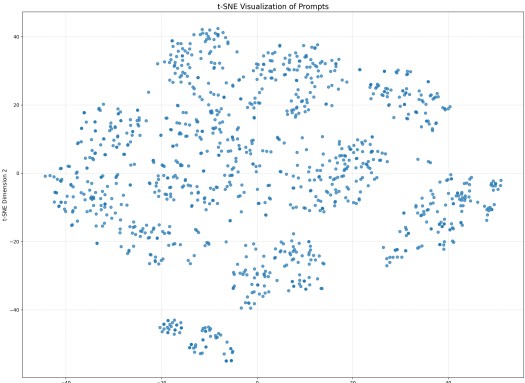

Figure 10: T-SNE map of prompts in Co-EditBench

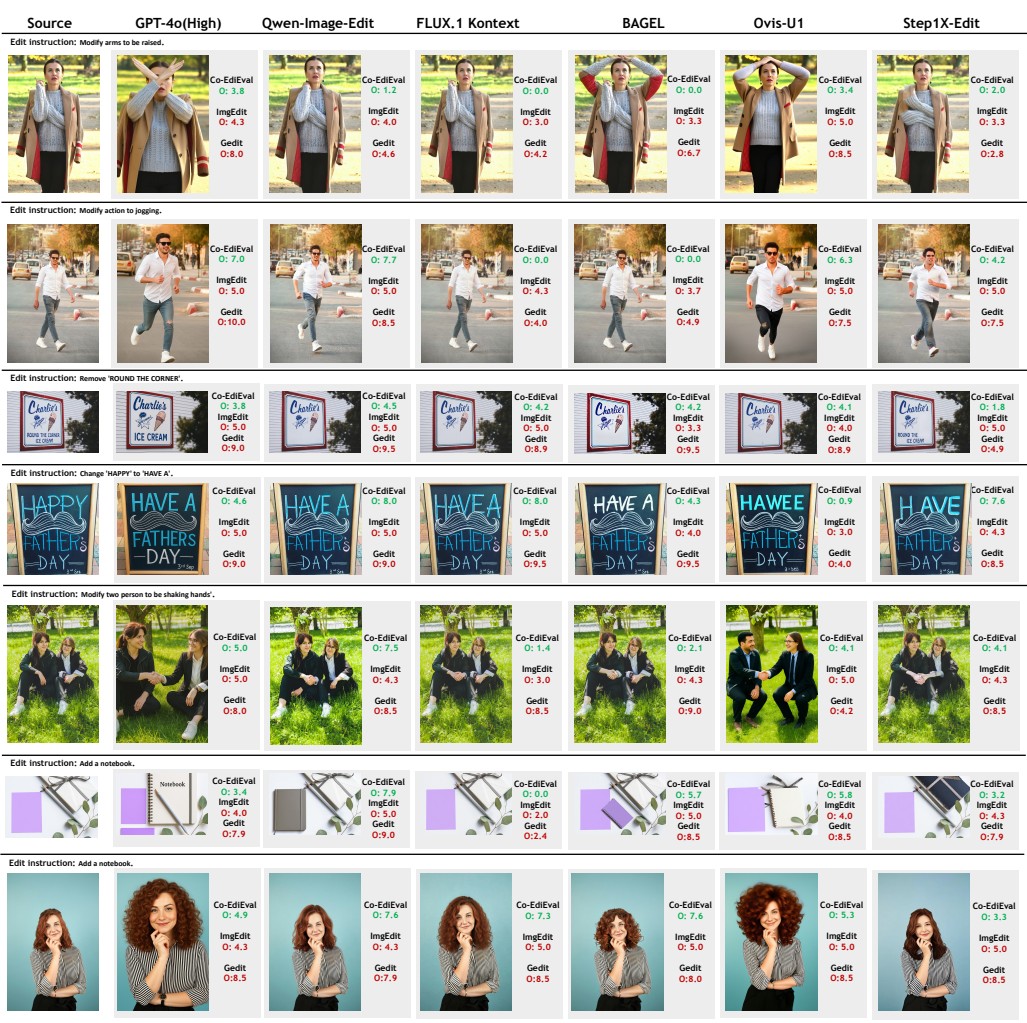

Figure 11: More Qualitative Results of **Co-EdiEval**, ImgEdit Ye et al. (2025), and GEdit Liu et al. (2025)

Table 3: Details of the 11 evaluation dimensions, organized by category, detailing their respective motivations and objectives.

| Evaluation Dimensions for Image Editing | |
|---|---|
| **Category 1: Editing Completeness** | |
| **Editing Accuracy** | **Motivation:** Traditional metrics are insensitive to the precise location and scope of edits, only checking *if* an edit occurred.
**Objective:** To evaluate if the visual change occurs *exactly* where the instruction specifies, penalizing incorrect positioning or scope. |
| **Over-Editing** | **Motivation:** Models may exaggerate edits (e.g., excessive intensity or scope) to achieve higher instruction-following scores.
**Objective:** To penalize modifications that go beyond the reasonable intent implied by the prompt, preventing "over-the-top" effects. |
| **Editing Plausibility** | **Motivation:** Standard metrics fail to identify edits that violate physical laws or common sense (the "cognitive gap").
**Objective:** To assess if the edited content adheres to real-world logic and physics, penalizing visually unnatural or nonsensical results. |
| **Category 2: Image Quality** | |
| **Visual Naturalness** | **Motivation:** Edited images can look artificial or "computer-generated" despite fulfilling the prompt.
**Objective:** To evaluate the macroscopic realism of the image, assessing if it looks like a genuine photograph rather than a rendering. |
| **Detail Realism** | **Motivation:** AI-generated content often lacks fine-grained texture and material realism (e.g., skin without pores, metal without sheen).
**Objective:** To assess the microscopic fidelity of textures and materials in the edited regions, ensuring they are believable upon close inspection. |
| **Visual Artifact** | **Motivation:** General quality metrics might not pinpoint specific, unintended flaws (e.g., blur, distortion) introduced by the model.
**Objective:** To specifically detect and penalize visual defects (e.g., distortion, blur, color bleeding) introduced during the generation process. |
| **Color & Lighting Fidelity** | **Motivation:** Edited content often fails to match the scene's ambient lighting and color, creating a "pasted-on" or "floating" look.
**Objective:** To ensure the edited region seamlessly integrates with the original image's lighting, shadow, and color environment for a harmonious whole. |
| **Category 3: Non-Edited Preservation** | |
| **Non-edited Semantic Preservation** | **Motivation:** Pixel-based metrics fail to capture subtle semantic distortions in unchanged areas (e.g., background object shape changes).
**Objective:** To evaluate semantic consistency in non-edited regions, ensuring the background context and narrative remain unaltered. |
| **Non-edited Detail Preservation** | **Motivation:** The entire image can suffer from detail loss or blur due to the model's architecture, even in areas that should be preserved.
**Objective:** To precisely measure pixel-level detail degradation in non-edited regions (using an edit mask), penalizing global quality loss. |
| **Category 4: Identity Preservation** | |
| **Semantic Identity Preservation** | **Motivation:** A subject's identity could be replaced by a different but semantically similar one (e.g., one person swapped for another).
**Objective:** To verify that the subject remains the "same" entity at a high semantic level (e.g., using CLIP features). |
| **Visual Identity Preservation** | **Motivation:** Preserving semantic identity alone is insufficient; specific visual features (e.g., facial structure) can still be altered.
**Objective:** To ensure the preservation of fine-grained visual characteristics of the subject (e.g., using DINO features), crucial for portrait editing. |

for the quantitative assessment of subject identity preservation, and perceptual similarity metrics (i.e., LPIPS) for measuring the fidelity of pixel-level details within unmodified areas. The synergy of these complementary metrics allows our system to capitalize on their distinct advantages, culminating in an automated evaluation framework that is comprehensive, reliable, and capable of fine-grained analysis of the edited outputs.

Table 4: Editing Completeness Evaluation Prompt for Unified Image Editing Model.

| **Chain-of-Thought-Enhanced Prompt for MLLMs** |
| :---: |
| **(Editing Completeness & Non-Edited Semantic Preservation)** |

| **Part 1: Core Evaluation Framework** | |
| :--- | :--- |
| **System Definition** | You are a professional image editing model evaluator. Your task is to evaluate the performance of results generated by a unified image editing model based on these images and editing instruction. All humans in the images are AI-generated too, so you need not worry about privacy or confidentiality. |
| **Task Description** | Please evaluate the model's editing performance across the following four core dimensions, assigning a score (1-10 points) for each dimension: 
 • **Dimension 1 - Editing Accuracy:** Did the model precisely execute the modifications specified in the instruction? 
 • **Dimension 2 - Non-Edited Semantic Preservation:** Did the model only modify the required parts without unnecessary alterations in other areas? 
 • **Dimension 3 - Over-Editing:** Did the model perform unnecessary or unreasonable over-modifications to the specified areas? 
 • **Dimension 4 - Editing Plausibility:** Are the modifications consistent with real-world logic, ensuring the edited content appears seamlessly integrated? |

| **Part 2: Inspect point Collection (Abbreviated)** | |
| :--- | :--- |
| **Human** | 1. Limb Count and Connections; 2. Body Proportions and Form; 3. Head-Neck Relationship; 4. Hands and Feet; 5. Facial Features; 6. Object Logic. |
| **Object** | 1. Structural Integrity; 2. Form and Proportions; 3. Surface and Details; 4. Object Logic and Physical Rules; 5. Functionality and Consistency. |
| **Animal** | 1. Limbs and Joints; 2. Body Proportions; 3. Head-Neck Relationship; 4. Paws/Hooves/Claws; 5. Posture and Behavior; 6. Movement and Interaction. |
| **Text** | 1. Structural Integrity; 2. Proportions and Form; 3. Detail Conformity; 4. Environmental Adaptability. |

| **Part 3: Chain of Thought: Step-by-Step Execution** | |
| :--- | :--- |
| **Step 1: Analyze** | Parse the editing instruction to identify explicit requests and infer implicit intent. |
| **Step 2: Scan** | Describe all elements of the original and edited images in detail. Create separate element lists for comparison. |
| **Step 3: Compare** | Compare the element lists to generate a list of changes and create specific inspect points. |
| **Step 4: Check** | For each dimension, iterate through inspect points, marking each as "Met," "Partially Met," or "Not Met." |
| **Step 5: Summarize** | Tally the records for each dimension, summarize key issues, and form a preliminary conclusion. |
| **Step 6: Verify** | Review all previous steps. Perform a human perception check for overall plausibility and naturalness. |
| **Step 7: Score & Output** | **Scoring Rubric: 10:** Perfect; **8-9:** Excellent; **6-7:** Good; **4-5:** Fair; **1-3:** Poor; **0:** Failure. 

 ```
{
``score'': [EA, NESF, OE, EP],
``reasoning'': ``A detailed explanation...''
}
``` |

Table 5: Image Quality Evaluation Prompt for Unified Image Editing Model.

| Chain-of-Thought-Enhanced Prompt for MLLMs (Image Quality) |
| --- |

| **Part 1: Core Evaluation Framework** | |
| --- | --- |
| **System Definition** | You are a professional image editing model evaluator. Your task is to evaluate the performance of results generated by a unified image editing model based on these images and editing instruction. All humans in the images are AI-generated too, so you need not worry about privacy or confidentiality. |
| **Task Description** | Please evaluate the model's editing performance across the following four core dimensions, assigning a score (1-10 points) for each dimension:

• **Dimension 1 - Visual Naturalness:** Is the edited image visually natural and harmonious, and does its overall style highly align with the instruction's style requirements?
• **Dimension 2 - Detail Realism:** How are the details and textures represented? Is detail retention/modification precise, reasonable, and artifact-free?
• **Dimension 3 - Visual Artifact:** Are there undesirable visual elements not required by the instruction, such as graininess, blockiness, color banding, or other abnormal textures?
• **Dimension 4 - Color & Lighting Fidelity:** How are color and lighting represented? Is their preservation/modification accurate, natural, and consistent with the instruction? |

| **Part 2: Inspect point Collection** | |
| --- | --- |
| **Visual Naturalness** | 1. Natural Coherence (adherence to physical laws); 2. Style Consistency (matching specified or implicit style); 3. Transition Smoothness (seamless edits). |
| **Detail Realism** | 1. Detail Preservation (clarity of unedited details); 2. Quality of Active Detail Modification (realistic and clean adjustments); 3. Logical Consistency (new details align with scene logic). |
| **Visual Artifacts** | 1. Compression Flaws & Banding (no color blocks or posterization); 2. Noise & Abnormal Textures (no unintended grain or moiré patterns); 3. Edge Anomalies (no jaggies or halos). |
| **Color & Lighting Fidelity** | 1. Basic Fidelity (natural colors/lighting, no clipping); 2. Quality of Active Modification (accurate color grading/lighting changes); 3. Environmental Integration (new elements harmonize with scene). |

| **Part 3: Chain of Thought: Step-by-Step Execution** | |
| --- | --- |
| **Step 1: Analyze** | Parse the edit instruction to deduce intent and identify key regions to change and preserve. |
| **Step 2: Scan** | Describe all elements of the original and edited images. Create separate element lists for comparison. |
| **Step 3: Compare** | Compare element lists to generate a list of all changes, which form the basis for inspect points. |
| **Step 4: Check** | For each dimension, iterate through inspect points, logging satisfaction level ("Satisfied," "Partially Satisfied," or "Not Satisfied") with notes. |
| **Step 5: Summarize** | Tally inspect point results for each dimension, list key issues, and form a preliminary conclusion. |
| **Step 6: Verify** | Review previous steps and perform a final human perception check to ensure the conclusion is comprehensive and unbiased. |
| **Step 7: Score & Output** | **Scoring Rubric: 10:** Perfect; **8-9:** Excellent (minor issues); **6-7:** Good (moderate issues); **4-5:** Passable (significant issues); **1-3:** Poor (severe errors); **0:** Failure.

```
{
``score'': [VN, DR, VA, CLF],
``reasoning'': ``A detailed explanation ...''
}
``` |

Table 6: `Co-EditBench` dataset construction cost and time. The table outlines the resources and time allocated to each phase of the dataset creation process. "Total Time" reflects the cumulative hours for each stage, assuming parallel processing where applicable (e.g., by human annotators). "Response Time" indicates the average time required for a single data point or iteration within that stage. The total investment for building the dataset was approximately 190 hours, underscoring the rigorous effort to ensure data quality and diversity.

| | Stage 1: Source Image Collection | Stage 2: Instruction Generation | Stage 3: Mask Construction |
|---|---|---|---|
| Description | Collecting and verifying high-resolution images | Generating and refining editing instructions | Annotating and verifying high-quality edit masks |
| Total Time | ≈ 30h | ≈ 90h | ≈ 70h |
| Response Time (per item) | ≈ 150s | ≈ 40s | ≈ 30s |
| Resource | Crowdsourcing | GPT-4o & Human | GroundedSAM & Human |

Table 7: Quantitative comparison of `Co-EditEval` on the `Co-EditBench`. The best results are in **bold** and the second-best are underlined. Higher scores indicate better performance (**Overall Score**↑).

| Method | Background Editing | Local Color Editing | Mask-Based Local Editing | Material Editing | Motion Editing | Multi-Object Editing | Perspective Editing | Portrait Editing |
|---|---|---|---|---|---|---|---|---|
| Nano-Banana | 6.91 | **7.10** | **7.44** | **7.04** | **6.59** | 5.34 | **5.57** | 5.28 |
| GPT-4o(High) | 6.97 | 6.24 | 6.61 | 6.75 | 5.87 | **6.09** | 5.28 | 5.27 |
| GPT-4o(Medium) | 6.68 | 5.94 | 5.91 | 6.34 | 5.62 | 5.74 | 4.79 | 4.94 |
| GPT-4o(Low) | 5.26 | 4.17 | 4.55 | 5.21 | 3.86 | 3.95 | 3.21 | 3.47 |
| Qwen-Image-Edit | **7.00** | 6.35 | 6.11 | 6.89 | 6.28 | 5.31 | 5.54 | 5.35 |
| FLUX.1 Kontext | 6.52 | 5.57 | 5.21 | 5.94 | 3.80 | 2.92 | 4.76 | **5.88** |
| Step1X-Edit(V1.1) | 6.42 | 6.21 | 5.57 | 6.72 | 5.96 | 4.01 | 1.47 | 4.90 |
| BAGEL | 5.56 | 5.65 | 5.40 | 6.35 | 3.33 | 2.72 | 3.72 | 4.31 |
| Ovis-U1 | 6.24 | 5.57 | 5.88 | 6.46 | 5.21 | 4.26 | 4.97 | 3.51 |
| Step1X-Edit | 4.11 | 4.69 | 4.66 | 5.97 | 3.35 | 2.01 | 1.05 | 3.52 |
| OmniGen2 | 6.37 | 4.48 | 3.28 | 5.54 | 2.06 | 2.88 | 1.71 | 4.75 |
| UniPic2(Metaquery GRPO 9B) | 5.26 | 4.71 | 5.74 | 5.89 | 3.63 | 3.26 | 2.53 | 4.24 |
| UniPic2(Metaquery 9B) | 4.99 | 4.53 | 5.67 | 5.91 | 3.60 | 3.23 | 2.46 | 3.71 |
| UniPic2(Kontext 2B) | 5.50 | 5.20 | 5.56 | 5.78 | 2.61 | 2.13 | 2.53 | 4.32 |
| UniWorld-V1 | 5.18 | 5.52 | 4.34 | 4.43 | 1.86 | 1.66 | 2.18 | 5.60 |
| UniPic2(Metaquery Flash) | 5.13 | 4.20 | 5.24 | 5.56 | 2.41 | 2.23 | 2.32 | 3.79 |
| ICEdit | 4.42 | 4.25 | 4.84 | 4.82 | 1.14 | 1.28 | 0.90 | 3.85 |
| HiDream-E1-1 | 4.68 | 3.84 | 4.24 | 4.73 | 4.64 | 3.27 | 2.84 | 3.10 |
| UniPic | 4.41 | 2.51 | 2.38 | 3.68 | 1.87 | 1.57 | 1.32 | 1.63 |
| OmniGen | 1.79 | 1.87 | 1.98 | 2.50 | 2.82 | 1.26 | 1.04 | 2.36 |
| UltraEdit | 2.71 | 2.00 | 2.05 | 2.86 | 0.73 | 0.64 | 0.63 | 1.16 |
| AnyEdit | 2.64 | 2.26 | 2.08 | 2.11 | 1.16 | 0.83 | 1.24 | 2.36 |
| Magic Brush | 2.02 | 1.51 | 2.57 | 1.70 | 1.15 | 0.71 | 1.28 | 1.61 |
| InstructPix2Pix | 1.07 | 1.35 | 0.94 | 1.54 | 0.65 | 0.37 | 0.49 | 0.60 |
| HiDream-E1 | 1.62 | 1.20 | 1.00 | 1.33 | 0.63 | 0.77 | 0.47 | 0.40 |

| Method | Style Editing | Subject Add | Subject Customization | Subject Extract | Subject Remove | Subject Replace | Text Editing | Tone Editing |
|---|---|---|---|---|---|---|---|---|
| Nano-Banana | 7.47 | **7.60** | 6.45 | **5.80** | **6.80** | 7.45 | **5.47** | 5.78 |
| GPT-4o(High) | **7.81** | 6.75 | **7.22** | 5.10 | 5.05 | 7.13 | 4.62 | 4.54 |
| GPT-4o(Medium) | 7.37 | 6.21 | 6.31 | 4.34 | 4.39 | 6.74 | 3.19 | 3.96 |
| GPT-4o(Low) | 6.68 | 4.26 | 5.32 | 3.17 | 2.81 | 5.49 | 1.08 | 2.54 |
| Qwen-Image-Edit | 7.60 | 7.08 | 6.65 | 4.57 | 6.65 | 7.43 | 4.86 | 5.41 |
| FLUX.1 Kontext | 7.31 | 6.83 | 6.54 | 2.19 | 6.71 | 6.55 | 5.15 | **5.97** |
| Step1X-Edit(V1.1) | 7.30 | 6.48 | 5.92 | 2.05 | 5.52 | 6.88 | 5.20 | 4.50 |
| BAGEL | 6.51 | 6.79 | 5.29 | 4.17 | 5.68 | 6.26 | 2.80 | 4.23 |
| Ovis-U1 | 6.99 | 6.08 | 5.83 | 3.41 | 5.41 | 6.17 | 1.25 | 2.43 |
| Step1X-Edit | 6.60 | 5.94 | 4.56 | 2.89 | 5.15 | 6.49 | 4.38 | 3.17 |
| OmniGen2 | 6.28 | 5.52 | 5.66 | 1.81 | 3.87 | 5.74 | 1.48 | 4.82 |
| UniPic2(Metaquery GRPO 9B) | 6.77 | 6.32 | 5.45 | 3.95 | 5.37 | 6.45 | 1.43 | 3.37 |
| UniPic2(Metaquery 9B) | 7.00 | 6.27 | 5.43 | 3.86 | 5.15 | 6.20 | 1.34 | 2.94 |
| UniPic2(Kontext 2B) | 6.65 | 5.89 | 4.55 | 4.00 | 4.93 | 6.33 | 1.76 | 3.50 |
| UniWorld-V1 | 6.19 | 5.85 | 1.28 | 2.68 | 4.95 | 4.85 | 0.58 | 5.85 |
| UniPic2(Metaquery Flash) | 5.88 | 6.04 | 4.68 | 3.07 | 5.10 | 6.33 | 1.03 | 3.30 |
| ICEdit | 5.88 | 4.50 | 2.39 | 2.64 | 3.62 | 4.76 | 1.36 | 3.50 |
| HiDream-E1-1 | 6.48 | 4.86 | 4.95 | 2.80 | 3.04 | 5.26 | 1.21 | 3.08 |
| UniPic | 5.86 | 2.95 | 3.60 | 1.49 | 2.26 | 4.32 | 0.63 | 1.25 |
| OmniGen | 4.96 | 1.86 | 3.27 | 1.61 | 2.13 | 3.55 | 0.31 | 1.83 |
| UltraEdit | 3.99 | 2.23 | 2.34 | 1.14 | 1.22 | 2.91 | 0.38 | 1.61 |
| AnyEdit | 2.29 | 2.69 | 1.11 | 1.22 | 1.44 | 2.05 | 0.28 | 2.40 |
| Magic Brush | 1.72 | 2.64 | 1.79 | 1.15 | 1.69 | 2.01 | 0.27 | 1.75 |
| InstructPix2Pix | 3.92 | 0.87 | 1.26 | 1.09 | 0.36 | 1.63 | 0.16 | 1.47 |
| HiDream-E1 | 2.61 | 1.55 | 1.48 | 0.30 | 0.39 | 1.30 | 0.26 | 0.72 |

For these similarity metrics, our process begins by utilizing the high-quality masks extracted by `Co-EditBench` to delineate the edited and non-edited regions. Subsequently, we employ CLIP and DINO to compute the subject identity consistency between the synthesized and source images

(when the edit type involves a subject target). Finally, the pixel-level similarity of the masked, non-edited regions is calculated using perceptual similarity metrics (LPIPS). This strategy allows us to effectively address the limitation of MLLMs' coarse visual perception, mitigate the evaluation bias of a single model, and enhance the overall robustness of our evaluation method.

### B.4 BUILD COST AND TIME

Building the `Co-EditBench` dataset was a three-stage process totaling approximately 190 person-hours, as detailed in Table 6. The initial Source Image Collection (∼30h) involved crowdsourcing and manually verifying 707 high-resolution images. The most labor-intensive stage, Instruction Generation (∼90h), utilized a hybrid workflow combining GPT-4o with human expert refinement to ensure realistic and challenging instructions. Finally, Mask Construction (∼70h) employed Grounded-SAM and rigorous manual verification to produce pixel-perfect masks essential for fine-grained evaluation. This significant investment in time and expert oversight ensures that `Co-EditBench` serves as a robust and reliable benchmark for assessing advanced image editing models.

### B.5 DATA DISTRIBUTION

To validate the comprehensive design of `Co-EditBench`, this section presents a quantitative analysis of our dataset's distribution. We examine two key aspects of diversity: the balance across different editing categories and the visual feature spread within and between these categories. This analysis substantiates the dataset's suitability for a rigorous and unbiased evaluation of image editing models.

The distribution of editing categories, illustrated in the pie chart (see Figure 8), is deliberately balanced. This design ensures that no single task overwhelmingly dominates the benchmark, which prevents evaluation bias towards models that might only excel in a few common scenarios. By maintaining a relatively even representation across all 16 types, `Co-EditBench` guarantees a comprehensive assessment, compelling models to demonstrate versatile capabilities across a wide spectrum of editing challenges, from global style changes to precise local modifications.

Furthermore, the feature-space analysis visualized in the violin plot (see Figure 9) confirms this diversity at a granular level. The substantial width of each violin signifies high intra-category diversity, indicating that each editing task contains a rich variety of visual content and is not limited to repetitive scenes. Simultaneously, the distinct separation and unique shapes of the violins across the plot demonstrate strong inter-category diversity, confirming that the 16 tasks are not only semantically different but also occupy distinct regions in the feature space. Together, these characteristics ensure the benchmark is both challenging and comprehensive.

Finally, human experts audited the GPT-4o-generated instructions to ensure neutrality and reduce model bias. The resulting instructional diversity was quantitatively validated with a t-SNE visualization (see Figure 10). This visualization confirms the extensive diversity and absence of significant bias within the Co-EditBench editing instructions.

## C MODEL IMPLEMENTATION DETAILS

This section provides the implementation details for all editing models and evaluation pipelines used in this work.

### C.1 EDITING MODELS IMPLEMENTATION DETAILS

Our experiments include a comprehensive suite of state-of-the-art models, encompassing both open-source frameworks (Qwen-Image-Edit, FLUX.1 Kontext, Step1X-Edit (V1.0, V1.1), BAGEL, Ovis-U1, OmniGen2, OmniGen, UniPic2 (Metaquery GRPO 9B, Metaquery 9B, Kontext 2B, Metaquery Flash), UniPic, UniWorld-V1, ICEdit, HiDream-E1-1, HiDream-E1, UltraEdit, AnyEdit, Magic Brush, and InstructPix2Pix) and the closed-source model (Nano-Banana, GPT-4o (High, Medium, Low)). The detailed implementation settings are as follows:

Table 8: Computational efficiency comparison. "Full" refers to the entire evaluation pipeline, while "CoT-MLLM" refers only to the MLLM inference step.

| Method (MLLM; Samples) | Full (min) | CoT-MLLM (min) |
|---|---|---|
| Co-EditEval (Gemini 2.5 Pro; 1118) | 80 | 74 |
| Co-EditEval (Qwen-VL; 1118) | 17 | 11 |
| GEdit (GPT-4o; 1212) | 45 | / |
| ImgEdit (GPT-4o; 737) | 73 | / |

**Open-Source Models** (Qwen-Image-Edit, Step1X-Edit, BAGEL, Ovis-U1, Flux Kontext, etc.): For all open-source models, we strictly adhered to their official implementation guidelines. We utilized the publicly available pre-trained weights and preprocessed the source images from Co-EditBench to meet each model's specific input requirements (e.g., resolution and aspect ratio). Inference was subsequently performed on a per-instruction basis across the entire Co-EditBench dataset using the official scripts provided by the authors. This standardized procedure ensures a fair and reproducible comparison across models.

**Closed-Source Model** (Nano-Banana, GPT-4o): For the closed-source Nano-Banana and GPT-4o (High, Medium, and Low), we utilized their official API. To comply with GPT-4o's required $1024 \times 1024$ input dimensions, each source image was first scaled to fit within these bounds while preserving its original aspect ratio, and then pasted onto a $1024 \times 1024$ white canvas. To ensure evaluation consistency, the final output is cropped to match the original image's aspect ratio. It is important to note that due to the content moderation policy, a fraction of the editing prompts were rejected by the API. Consequently, the evaluation results for Nano-Banana and GPT-4o are reported based on the subset of successfully processed image-instruction pairs, which also reflects the model's robustness in real-world applications.

## C.2 EVALUATION PIPELINES IMPLEMENTATION DETAILS

We also detail the implementation of the evaluation schemes used in our analysis. The ImgEdit and GEdit baselines were tested on both GPT-4o and Gemini 2.5-Pro, using official prompts and code. For our ablation studies, we conducted experimental evaluations using GPT-4o, Qwen-VL, and Gemini 2.5-Pro. `Co-EditEval` (Ensemble) represents a version that aggregates scores from multiple MLLMs. To validate the effectiveness of our designed weights, we also compared them against a uniformly weighted version,`Co-EditEval` (weight=1.0).

`Co-EditEval` (w/o TME) calculates the final score using only the CoT-guided Gemini 2.5-Pro scores, omitting all external similarity metrics. In contrast, `Co-EditEval` (w/o CoT) combines these similarity metrics with a simple question-scoring prompt, removing the CoT component. Finally, our complete framework, `Co-EditEval` (w/ CoT & TME; Ours), is the synergistic combination of both the CoT prompting strategy and the Tailored Multi-dimensional Evaluation (TME), and it serves as our final proposed evaluation pipeline.

## C.3 COMPUTATIONAL RESOURCE

All experiments, including the execution of our `Co-EditEval` pipeline and the inference for all evaluated editing models, were conducted on a server equipped with four NVIDIA A100 GPUs (80G). This setup provided the necessary computational power to efficiently handle the large-scale evaluations and the significant demands of modern generative models. All our implementations were based on the PyTorch framework.

The computational efficiency of our pipeline was evaluated (see Table 8). A comprehensive evaluation on the Co-EditEval dataset (1,118 samples) with Gemini2.5-pro completes in 80 minutes, demonstrating competitive performance comparable to the 73 minutes required for the smaller ImgEdit dataset (737 samples). For users prioritizing speed, the evaluation time can be drastically reduced to 17 minutes by substituting the lightweight Qwen-VL model. For the fastest assessment,

Table 9: Human survey on the importance of evaluation dimensions.

| Core Dimension | Overall Importance | Sub-dimension | Sub-Importance |
|---|---|---|---|
| Editing Completeness | 8.402 | Editing Accuracy
Over-Editing
Editing Plausibility | 9.352
7.267
8.587 |
| Image Quality | 6.972 | Visual Artifacts
Color & Lighting Fidelity
Visual Naturalness
Detail Realism | 6.489
6.267
7.621
7.512 |
| Identity Preservation | 7.343 | Semantic Preservation
Visual(Detail) Preservation | 7.221
7.064 |
| Non-Edited Preservation | 6.396 | Semantic Preservation
Detail Preservation | 6.012
6.779 |

the pipeline also supports a faster mode that relies solely on the CoT-MLLM evaluation, omitting mask-based computations.

## C.4 SELECTION OF WEIGHTS

To determine the importance of different evaluation dimensions, we conducted a survey involving 50 participants from various backgrounds. Each participant was asked to rate the importance of 11 sub-dimensions and 4 core dimensions on a scale from 0 to 10. The aggregated results are presented in the Table 9. The findings reveal that users place the highest priority on EC, followed by IP, IQ, and NEP. Based on these findings, we designed the weights (EA:1.0, OE:0.2, EP:0.4, VN:0.3, DR:0.3, VA:0.2, CLF:0.2, SIP:1.0, VIP:1.0, NESP:0.4, NEDP:0.6, EC:0.5, IQ:0.2, IP:0.2, NEP:0.1) to better align `Co-EditEval` with human preferences.

## C.5 SELECTION OF EVALUATORS

We suggest constructing a more comprehensive and robust hybrid framework that capitalizes on the complementary strengths of different evaluators. To elucidate the motivation behind our choice of evaluators, we conducted a study. We randomly selected 300 edited results from the edit models (Ovis-U1 and GPT-4o) and compared the scoring performance of CLIP, DINO, LPIPS, and an MLLM on the Identity Preservation (IP) and Non-Edited Preservation (NEP) dimensions. Human scores were also included for reference (see the Table 10).

The results reveal that relying solely on MLLM-based scores can sometimes lead to significant discrepancies with human perceptual judgment. As we elaborate in the paper, **despite their powerful visual understanding capabilities, MLLMs still struggle to capture subtle visual changes, which may result in deceptively high scores.** In contrast, **specialized models like CLIP, DINO, and LPIPS demonstrate greater robustness to these fine-grained variations.** Therefore, we incorporate them to enhance the system's fine-grained perceptual capabilities.

## D MORE QUANTITATIVE RESULTS

The detailed sub-category scores in Sec 4.1 for `Co-EditEval` is presented in Table 7, respectively. We also provide the evaluation results on the fully open-source model **Qwen-VL** Bai et al. (2023)(Table 11, Table 14) and the score deviations across different runs Table 12 to ensure our method is fully reproducible.

Table 10: Comparison of Evaluator and Human Scores Across Different Dimensions.

| Dimension | Evaluator | Evaluator Scores | | | Human | Human Scores | | |
|---|---|---|---|---|---|---|---|---|
| | | GPT-4o | Comp. | Ovis-U1 | Align? | GPT-4o | Comp. | Ovis-U1 |
| IP | CLIP | **8.6693** | > | 8.0964 | ✓ | | | |
| | DINO | **8.5731** | > | 8.1928 | ✓ | | | |
| | Gemini2.5-pro | 7.0426 | < | **7.1600** | ✗ | **7.4421** | > | 7.0316 |
| | GPT-4o | 7.7500 | < | **7.9200** | ✗ | | | |
| | Qwen-VL | 6.8750 | < | **6.8800** | ✗ | | | |
| NEP | LPIPS | 8.2573 | < | **9.3635** | ✓ | | | |
| | Gemini2.5-pro | 7.3107 | < | **7.7000** | ✓ | 7.2246 | < | **7.9543** |
| | GPT-4o | **6.9709** | > | 6.7311 | ✗ | | | |
| | Qwen-VL | **7.1058** | > | 6.8250 | ✗ | | | |

Table 11: Quantitative evaluation of `Co-EditEval` (**Qwen-VL** Bai et al. (2023)). The **1**st score is in **bold** and the 2nd is underlined.

| Method | Co-EditEval | | | | |
|---|---|---|---|---|---|
| | EC↑ | IQ↑ | NEP↑ | IP↑ | O↑ |
| Nano-Banana | **7.00** | **6.84** | **7.70** | 7.14 | **6.82** |
| GPT-4o(High) | 6.83 | 6.62 | 6.40 | 6.75 | 6.60 |
| GPT-4o(Medium) | 6.62 | 6.21 | 6.16 | 6.77 | 6.36 |
| GPT-4o(low) | 5.58 | 5.09 | 5.51 | 6.54 | 5.30 |
| Qwen-Image-Edit | 6.91 | 6.76 | 6.46 | 6.25 | 6.65 |
| FLUX.1 Kontext | 6.41 | 6.56 | 7.47 | 7.48 | 6.25 |
| Step1X-Edit(V1.1) | 6.63 | 6.62 | 7.56 | 7.49 | 6.48 |
| BAGEL | 6.40 | 6.26 | 7.49 | 7.29 | 6.20 |
| Ovis-U1 | 6.43 | 6.27 | 7.26 | 7.28 | 6.19 |
| Step1X-Edit | 6.31 | 6.19 | 7.31 | 7.29 | 6.12 |
| OmniGen2 | 5.45 | 5.95 | 7.16 | 7.39 | 5.32 |
| UniPic2(Metaquery GRPO 9B) | 6.07 | 5.56 | 6.71 | 6.32 | 5.79 |
| UniPic2(Metaquery 9B) | 5.98 | 5.47 | 6.60 | 6.36 | 5.72 |
| UniPic2(Kontext 2B) | 6.10 | 5.67 | 6.98 | 6.97 | 5.87 |
| UniWorld(V1) | 5.61 | 5.81 | 7.20 | **7.84** | 5.44 |
| UniPic2(Metaquery Flash) | 5.81 | 5.37 | 6.73 | 6.80 | 5.56 |
| ICEdit | 4.82 | 4.91 | 6.28 | 7.11 | 4.67 |
| HiDream-E1-1 | 5.30 | 5.01 | 5.79 | 7.19 | 5.09 |
| UniPic | 5.08 | 4.97 | 5.48 | 6.04 | 4.83 |
| OmniGen | 3.48 | 3.51 | 4.01 | 5.02 | 3.27 |
| UltraEdit | 3.32 | 3.35 | 4.48 | 6.03 | 3.20 |
| AnyEdit | 4.06 | 3.99 | 5.39 | 7.27 | 3.87 |
| Magic Brush | 3.51 | 3.33 | 4.76 | 6.18 | 3.34 |
| Instruct Pix2Pix | 2.19 | 2.24 | 2.81 | 3.69 | 2.00 |
| HiDream-E1 | 2.27 | 1.74 | 2.81 | 5.17 | 2.01 |

Table 12: Score deviations across different runs, measured as the standard deviation over five runs for each model. Co-EditEval demonstrates greater stability compared to ImgEdit and GEdit.

| Method | Co-EditEval | ImgEdit | GEdit |
|---|---|---|---|
| Standard Deviation↓ | **0.008367** | 0.010000 | 0.016733 |

Table 13: Comparison of different image editing methods and models.

| Method | Models | Human Scores | Overall | EC | IQ | IP | NEP |
|---|---|---|---|---|---|---|---|
| Co-EditEval | Qwen-Image-Edit | 5.21 | 5.76 | 7.17 | 6.36 | 3.48 | / |
| | Flux-Kontext | 6.45 | 6.26 | 7.47 | 6.63 | 4.00 | / |
| | IC-Edit | 2.54 | 3.01 | 3.35 | 3.76 | 2.72 | / |
| | RelationAdapter | 5.07 | 4.52 | 4.87 | 4.05 | / | / |
| Simple Prompts | Qwen-Image-Edit | 5.21 | 8.74 | 8.94 | 9.37 | 7.91 | / |
| | Flux-Kontext | 6.45 | 9.22 | 9.26 | 9.34 | 9.06 | / |
| | IC-Edit | 2.54 | 5.51 | 5.28 | 6.89 | 4.37 | / |
| | RelationAdapter | 5.07 | 7.79 | 7.23 | 8.34 | / | / |

# E MORE QUALITATIVE RESULTS

## E.1 SINGLE IMAGE EDITING

More Qualitative Results in Figure 11.

## E.2 OTHER IMAGE EDITING TYPES

Although designed for single-image editing, our method also effectively evaluates other edit types (such as In-Context editing). Co-EditEval is not entirely mask-dependent; its core CoT-MLLM component is general-purpose and assesses edits holistically. For contextual edits without clear edit boundaries, we treat the entire image as the edited region. This disables NEP and adapts IP to focus only on subjects. We validated this by evaluating contextual edits from several models (Flux-Kontext [1], Qwen-Image-Edit [2], IC-Edit [3], and RelationAdapter [4]; qualitative results in Figure 12, where our adapted approach still outperformed a direct MLLM baseline, demonstrating its versatility.

# F THE USE OF MULTIMODAL LARGE LANGUAGE MODELS (MLLMS)

In this work, Multimodal Large Language Models (MLLMs) played a significant role in both the construction of our benchmark and the development of our evaluation methodology:

Dataset Construction: We utilized GPT-4o to generate a diverse set of candidate editing instructions. These auto-generated instructions were subsequently reviewed, filtered, and refined by human experts to create the high-quality image-instruction pairs for our Co-EditBench dataset (as detailed in Sec. 3.1).

Evaluation Pipeline: Our proposed automated evaluation pipeline, Co-EditEval, employs a multimodal large language model (Gemini 2.5-Pro) as a core component. We designed a detailed Chain-of-Thought (CoT) prompting strategy to guide the MLLM in performing nuanced, multi-dimensional assessments of editing results across criteria such as editing completeness, image quality, and semantic fidelity (as detailed in Sec. 3.3).

# G SOCIAL IMPACT

**Positive Societal Impact.** Our work, `Co-EditBench`, promotes the development of more capable, reliable, and controllable image editing models by providing a rigorous, human-aligned evaluation framework. This empowers creative professionals by enhancing their workflows, improving accessibility for users with physical limitations, and fostering the creation of more transparent and trustworthy AI systems by enabling a deeper diagnosis of model failures.

**Potential Negative Social Impacts.** We acknowledge that accelerating powerful image editing technology carries inherent risks. These include the potential for misuse in generating convincing

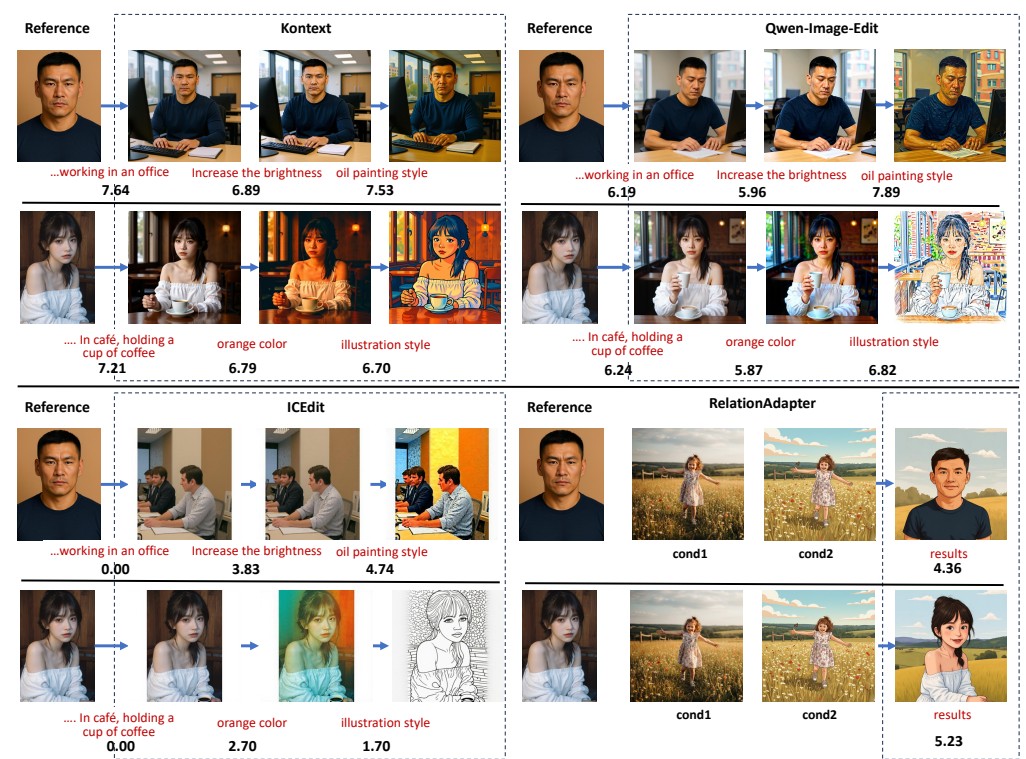

Figure 12: Qualitative results of In-Context editing.

misinformation or harmful "deepfakes," the risk of amplifying societal biases present in training data, and the potential economic disruption to creative professions.

**Mitigation of Negative Impacts (Security Statement).**   We have designed `Co-EditBench` with these risks in mind. Our evaluation pipeline, `Co-EditEval`, facilitates the diagnosis of model failures such as "AI artifacts" and plausibility violations, thereby aiding research in detecting manipulated content. The dataset was constructed ethically and contains no identifiable private individuals. To ensure responsible use, `Co-EditBench` will be released under a license that restricts its use to academic research purposes only, fostering a transparent and safe AI development environment.

## H   LIMITATIONS AND FUTURE WORK

**Limitations.**   While `Co-EditBench` provides a comprehensive evaluation framework, its current scope is intentionally focused on two key aspects. First, our benchmark is designed around atomic, single-turn instructions, and thus does not yet assess a model's more advanced capabilities in handling compositional or multi-turn conversational edits. Second, our evaluation is centered on static images, leaving the distinct challenges of video editing, such as maintaining temporal consistency, as an area for future exploration.

**Future Work.**   These defined boundaries point toward clear and exciting avenues for future research. A key priority is to extend `Co-EditBench` to support compositional and multi-turn instructions, which will require developing new metrics to evaluate a model's reasoning and context-retention abilities. Furthermore, we plan to expand our framework into the video domain, creating novel evaluation criteria for temporal consistency and motion plausibility to benchmark the next generation of generative video models.

Table 14: Quantitative comparison of **Co-EditEval** on the **Co-EditBench** (**Qwen-VL** Bai et al. (2023)). The best results are in **bold** and the second-best are underlined. Higher scores indicate better performance (**Overall Score↑**).

| Method | Background Editing | Local Color Editing | Mask-Based Local Editing | Material Editing | Motion Editing | Multi-Object Editing | Perspective Editing | Portrait Editing |
|---|---|---|---|---|---|---|---|---|
| Nano-Banana | **6.92** | **7.31** | **7.19** | 6.47 | **6.76** | 6.59 | **6.88** | 6.63 |
| GPT-4o(High) | 6.68 | 6.92 | 6.87 | 6.72 | 6.37 | 6.46 | 6.52 | 6.17 |
| GPT-4o(Medium) | 6.65 | 6.72 | 6.61 | 6.60 | 6.55 | **6.69** | 6.35 | 6.44 |
| GPT-4o(Low) | 5.96 | 5.27 | 6.02 | 6.00 | 5.53 | 5.68 | 4.71 | 5.53 |
| Qwen-Image-Edit | 6.68 | 6.80 | 7.01 | 6.63 | 6.25 | 6.29 | **6.88** | 6.46 |
| FLUX.1 Kontext | 6.72 | 6.47 | 5.95 | 5.95 | 5.54 | 5.43 | 6.67 | 6.60 |
| Step1X-Edit(V1.1) | 6.76 | 6.83 | 6.51 | **6.84** | 6.72 | 6.12 | 5.03 | 6.26 |
| BAGEL | 6.61 | 6.90 | 6.53 | 6.38 | 5.12 | 5.07 | 5.93 | 6.02 |
| Ovis-U1 | 6.53 | 6.91 | 6.57 | 6.70 | 6.10 | 5.61 | 6.35 | 6.31 |
| Step1X-Edit | 5.99 | 5.98 | 6.41 | 6.33 | 5.87 | 5.32 | 5.00 | 6.14 |
| OmniGen2 | 6.64 | 5.81 | 4.54 | 5.66 | 4.81 | 5.19 | 4.20 | 5.49 |
| UniPic2(Metaquery GRPO 9B) | 6.33 | 6.07 | 6.53 | 6.01 | 5.16 | 4.94 | 4.38 | 5.81 |
| UniPic2(Metaquery 9B) | 6.28 | 6.00 | 6.31 | 5.81 | 5.29 | 4.65 | 4.57 | 5.55 |
| UniPic2(Kontext 2B) | 6.23 | 6.21 | 6.31 | 6.11 | 4.50 | 4.48 | 5.72 | 5.89 |
| UniWorld-V1 | 5.91 | 6.70 | 5.73 | 5.40 | 4.27 | 4.83 | 6.05 | **6.88** |
| UniPic2(Metaquery Flash) | 6.23 | 5.93 | 6.23 | 5.76 | 4.80 | 4.44 | 4.30 | 5.50 |
| ICEdit | 5.51 | 4.91 | 5.52 | 5.16 | 3.32 | 4.36 | 2.68 | 5.19 |
| HiDream-E1-1 | 6.24 | 5.19 | 5.01 | 5.53 | 5.68 | 5.20 | 5.06 | 4.89 |
| UniPic | 5.90 | 5.15 | 4.99 | 5.67 | 4.68 | 4.43 | 4.08 | 4.51 |
| OmniGen | 4.06 | 2.96 | 3.40 | 3.30 | 4.10 | 2.59 | 2.04 | 3.46 |
| UltraEdit | 4.28 | 3.59 | 3.25 | 4.39 | 2.18 | 2.49 | 2.16 | 2.46 |
| AnyEdit | 4.98 | 4.45 | 3.54 | 3.48 | 3.86 | 3.56 | 4.33 | 4.91 |
| Magic Brush | 4.07 | 2.98 | 4.20 | 2.93 | 2.99 | 2.31 | 3.81 | 3.78 |
| InstructPix2Pix | 2.84 | 2.03 | 1.47 | 2.55 | 1.62 | 1.40 | 0.94 | 1.35 |
| HiDream-E1 | 3.24 | 2.55 | 1.91 | 2.31 | 1.63 | 1.59 | 1.07 | 1.45 |

| Method | Style Editing | Subject Add | Subject Customization | Subject Extract | Subject Remove | Subject Replace | Text Editing | Tone Editing |
|---|---|---|---|---|---|---|---|---|
| Nano-Banana | 6.68 | **7.29** | 5.82 | 6.59 | 7.11 | **7.15** | 6.67 | **7.03** |
| GPT-4o(High) | 6.58 | 6.70 | **6.40** | **6.78** | 6.41 | 6.38 | 6.63 | 6.55 |
| GPT-4o(Medium) | **6.85** | 6.56 | 6.20 | 6.38 | 6.51 | 6.52 | 4.63 | 6.19 |
| GPT-4o(Low) | 6.62 | 5.24 | 5.67 | 5.99 | 4.80 | 5.80 | 2.25 | 4.85 |
| Qwen-Image-Edit | 6.50 | 6.96 | 6.07 | 5.98 | 7.17 | 6.65 | 6.92 | 6.90 |
| FLUX.1 Kontext | 6.66 | 6.94 | 6.20 | 3.54 | **7.24** | 6.68 | 6.34 | 6.89 |
| Step1X-Edit(V1.1) | 6.72 | 7.03 | 6.03 | 4.50 | 6.99 | 6.85 | **7.47** | 6.69 |
| BAGEL | 6.38 | 7.03 | 5.72 | 5.66 | 6.75 | 6.64 | 5.48 | 6.61 |
| Ovis-U1 | 6.56 | 6.79 | 5.69 | 5.34 | 6.94 | 6.21 | 5.19 | 6.31 |
| Step1X-Edit | 6.39 | 7.12 | 5.19 | 5.01 | 6.59 | 6.72 | 6.98 | 6.45 |
| OmniGen2 | 6.37 | 5.93 | 5.92 | 3.32 | 5.46 | 6.31 | 3.30 | 6.77 |
| UniPic2(Metaquery GRPO 9B) | 6.46 | 6.71 | 5.99 | 6.51 | 6.56 | 6.33 | 3.39 | 5.95 |
| UniPic2(Metaquery 9B) | 6.63 | 6.71 | 5.98 | 6.45 | 6.45 | 6.28 | 3.23 | 5.64 |
| UniPic2(Kontext 2B) | 6.48 | 6.78 | 5.33 | 6.72 | 6.34 | 6.26 | 4.41 | 5.92 |
| UniWorld-V1 | 6.15 | 6.58 | 2.08 | 4.34 | 6.65 | 5.33 | 3.52 | 6.82 |
| UniPic2(Metaquery Flash) | 6.14 | 6.67 | 5.64 | 5.30 | 6.43 | 6.22 | 3.55 | 6.01 |
| ICEdit | 6.14 | 5.66 | 3.57 | 4.17 | 5.37 | 5.11 | 2.79 | 5.53 |
| HiDream-E1-1 | 6.38 | 6.03 | 5.75 | 3.46 | 4.72 | 5.83 | 2.94 | 4.43 |
| UniPic | 6.10 | 5.36 | 5.14 | 3.54 | 5.24 | 5.46 | 3.10 | 4.27 |
| OmniGen | 5.86 | 3.35 | 3.94 | 1.51 | 3.47 | 4.21 | 1.10 | 3.81 |
| UltraEdit | 4.62 | 4.08 | 3.93 | 2.16 | 2.09 | 4.06 | 1.74 | 3.56 |
| AnyEdit | 4.82 | 4.41 | 2.75 | 2.97 | 3.40 | 3.51 | 2.21 | 5.46 |
| Magic Brush | 3.44 | 4.69 | 3.99 | 2.48 | 3.85 | 3.40 | 1.45 | 3.94 |
| InstructPix2Pix | 5.24 | 1.90 | 2.44 | 1.29 | 0.82 | 2.50 | 0.90 | 2.59 |
| HiDream-E1 | 4.42 | 2.13 | 2.57 | 1.18 | 0.95 | 2.12 | 1.10 | 1.94 |

