# OpenReview forum: "Co-EditBench: Human-Aligned Benchmark for Instruction-Based Image Editing with Multi-Dimensional Assessment"
_ICLR.cc/2026/Conference — Submitted to ICLR 2026_

### Official Review · Reviewer_xYjs · 2025-10-29

**Soundness:** 2
**Presentation:** 3
**Contribution:** 3
**Rating:** 4
**Confidence:** 5

**Summary:**

This paper presents **Co-EditBench**, a benchmark for evaluating multimodal large language models (MLLMs) in instruction-guided image editing. It addresses limitations in existing benchmarks by providing a diagnostic dataset with 16 editing types, 11 fine-grained evaluation dimensions, and an automated pipeline (**Co-EditEval**) leveraging contextualized visual reasoning. Experiments show Co-EditBench aligns strongly with human judgment, offering more reliable and nuanced evaluations.

**Strengths:**

1. **Comprehensive Evaluation Benchmark**: Co-EditBench introduces a diagnostic dataset covering 16 real-world editing types and defines 11 fine-grained evaluation dimensions, addressing gaps in existing benchmarks and enabling nuanced assessments.

2. **Human-Aligned Evaluation**: The benchmark strongly correlates with human judgments, ensuring evaluations reflect subjective aesthetics and detailed visual perception.

**Weaknesses:**

1. **Incremental Evaluation Approach**: The evaluation method relies on weighted scoring based on existing feature similarity metrics and interpretable Chain-of-Thought reasoning. However, this approach feels incremental, as similar methodologies have been adopted by prior benchmarks like ImageEdit, Wise, and GenEval. The contribution in evaluation design is overstated and lacks substantial novelty.

2. **Regression to Outdated Metrics**: The inclusion of CLIP-based feature similarity metrics in the evaluation pipeline is concerning, as these metrics have been increasingly abandoned by recent works due to their lack of accuracy and reliability. While interpretable CoT-based evaluation is promising, relying on CLIP undermines the consistency and precision of the evaluation method.

3. **Missed Opportunity for RL-based Reward Models**: Instead of using feature-based similarity metrics, constructing RL-based reward models to validate multiple evaluation dimensions would provide a more scalable and interpretable solution. Such an approach would allow for checklist-based evaluations that are both flexible and robust, better aligned with current advancements in evaluation methodologies.

**Questions:**

The absence of RL-based reward models for explainability and verification in this work raises concerns about the contribution of its evaluation methodology. RL-based reward models provide a modern and promising approach for validating image edits by offering both scalability and interpretability. Their ability to dynamically adapt to evaluation criteria ensures alignment with human reasoning and increases robustness across diverse editing tasks.

---

> ### Author Response · Authors · 2025-11-19
> **Respond for Reviewer xYjs Part1**
>
> We are grateful for your meticulous evaluation and constructive suggestions. We appreciate the positive assessment of our dataset, experimental results, as well as the insightful questions raised. Our detailed point-by-point responses are provided below.
>
> **Response to W1:**
>
> We appreciate your suggestion. Our core innovation is not the simple integration of weighted scoring and MLLM-based evaluation, but a novel, systematic framework designed to address the "human perception gap" in image editing evaluation. The **table below** demonstrates this framework's significant advantage across four key aspects:
>
> **(1) Depth and Granularity of Evaluation:** In contrast to the three macro-level dimensions of ImgEdit, we propose 11 fine-grained dimensions and enhance the evaluation with Chain-of-Thought (CoT) reasoning. This enables a more precise and traceable diagnosis of failure modes.
>
> **(2)Task Specificity:** Our work is tailored for image editing, distinguishing it from benchmarks like WISE and GenEval, which focus on text-to-image (T2I) generation.
>
> **(3) Methodological Innovation:** We have developed a hybrid evaluation system. Unlike methods that rely solely on an MLLM (e.g., ImgEdit, WISE) or on rule-based detectors and classifiers (e.g., GenEval), our system synergizes a CoT-guided MLLM (semantics) with traditional metrics applied to precisely masked regions (details). This strikes a crucial balance between evaluation breadth and precision. Furthermore, our overall score calculation incorporates a completion-guided principle and a weighted geometric mean. This strategy penalizes imbalanced performance, discouraging models from "hacking scores" by excelling only in a few dimensions.
>
> **Experimental Validation:** Ablation studies (**paper-Table 2**) provide quantitative evidence that it is this systematic design that allows Co-EditEval to significantly surpass all baseline methods in its alignment with human judgment.
>
> | **Method**      | **Task**      | **Dimension** | **CoT** | **Region-aware** | **Paradigm**                 | **Analysis**                                             |
> | :-------------- | :------------ | :------------ | :------ | :--------------- | :--------------------------- | :------------------------------------------------------- |
> | **Co-EditEval** | Image Editing | 11sub+4core            | ✔       | ✔ (mask)         | Hybrid: MLLM (CoT) + Metrics | Deep, multi-dim framework; aligns with human perception. |
> | **ImgEdit**     | Image Editing | 3             | ✘       | ✘                | MLLM prompt scoring          | Coarse granularity; imprecise perception.                |
> | **WISE**        | Text-to-Image | 3             | ✘       | ✘                | MLLM weighted scoring        | Mismatched domain; simple weighting.                     |
> | **GenEval**     | Text-to-Image | 6             | ✘       | ✔ (bbox)         | Rule-based (detectors)       | Struggle to evaluate subjective aspects (e.g., style).   |
>
>
>
>
> | Method (Ablation Study)        |   SROCC↑   |   PLCC↑    |   KLCC↑    |   RMSE↓    |
> | :----------------------------- | :--------: | :--------: | :--------: | :--------: |
> | GEdit(GPT-4o)                  |   0.6492   |   0.6518   |   0.4889   |   2.8332   |
> | GEdit(Gemini2.5-Pro)           |   0.4414   |   0.4389   |   0.3248   |   3.1533   |
> | ImgEdit(GPT-4o)                |   0.5952   |   0.5850   |   0.4494   |   3.8820   |
> | ImgEdit(Gemini2.5-Pro)         |   0.7587   |   0.7653   |   0.6149   |   2.6032   |
> | evalname(Weights=1.0)          |   0.8397   |   0.8313   |   0.6560   |   2.4753   |
> | evalname (GPT-4o)              |   0.7226   |   0.6805   |   0.5436   |   2.4776   |
> | evalname (Qwen-VL)             |   0.7569   |   0.6966   |   0.5299   |   2.4900   |
> | evalname (Ensemble)            |   0.8253   |   0.7934   |   0.6552   |   2.2136   |
> | evalname(w/o TME)              |   0.8467   |   0.8304   |   0.6754   |   1.9389   |
> | evalname(w/o CoT)              |   0.8368   |   0.8451   |   0.6493   |   2.2198   |
> | **evalname(w/ CoT&TME; Ours)** | **0.8894** | **0.8657** | **0.7021** | **1.8132** |

---

> > ### Author Response · Authors · 2025-11-27
> > **Respond for Reviewer xYjs Part2**
> >
> > **Response to W2:**
> >
> > Thank you for your suggestion. We wish to clarify that our method does not advocate for a regression to traditional similarity metrics. **Instead, we suggest constructing a more comprehensive and robust hybrid framework that capitalizes on the complementary strengths of different evaluators.**
> > To elucidate the motivation behind our choice of evaluators, we conducted a study. We randomly selected 300 edited results from the edit models (Ovis-U1 and GPT-4o) and compared the scoring performance of CLIP, DINO, LPIPS, and an MLLM on the Identity Preservation (IP) and Non-Edited Preservation (NEP) dimensions. Human scores were also included for reference (see the **Table below**).
> >
> > | Dim  | Evaluator  | Eval. Score (GPT-4o vs Ovis-U1) | Human Score (GPT-4o vs Ovis-U1) | Align? |
> > | :--- | :--------- | :------------------------------ | :------------------------------ | :----- |
> > | IP   | CLIP       | **8.67** > 8.10                 | **7.44** > 7.03                 | ✓      |
> > |      | DINO       | **8.57** > 8.19                 |                                 | ✓      |
> > |      | Gemini-2.5 | 7.04 < **7.16**                 |                                 | ✗      |
> > |      | GPT-4o     | 7.75 < **7.92**                 |                                 | ✗      |
> > |      | Qwen-VL    | 6.88 < **6.88**                 |                                 | ✗      |
> > | NEP  | LPIPS      | 8.26 < **9.36**                 | 7.22 < **7.95**                 | ✓      |
> > |      | Gemini-2.5 | 7.31 < **7.70**                 |                                 | ✓      |
> > |      | GPT-4o     | **6.97** > 6.73                 |                                 | ✗      |
> > |      | Qwen-VL    | **7.11** > 6.83                 |                                 | ✗      |
> >
> > The results reveal that relying solely on MLLM-based scores can sometimes lead to significant discrepancies with human perceptual judgment. As we elaborate in the paper, **despite their powerful visual understanding capabilities, MLLMs still struggle to capture subtle visual changes, which may result in deceptively high scores.** In contrast, **specialized models like CLIP, DINO, and LPIPS demonstrate greater robustness to these fine-grained variations.** Therefore, we incorporate them to enhance the system's fine-grained perceptual capabilities.

---

> ### Author Response · Authors · 2025-11-27
> **Respond for Reviewer xYjs Part3**
>
> **Response to W3 and Q1:**
>
> Thank you for this profound insight. We acknowledge that RL models are a crucial technology for aligning human perception. Specifically, the "RL-based reward model" you mentioned shares common ground with the MLLMs we employ, as the latter are themselves complex reward models aligned through techniques like RLHF (e.g., GPT-4o[1], Qwen-VL[2]). Therefore, the choice we face is not "whether to use an RL model," but rather:
> **(A) Train a domain-specific reward model from scratch for the image editing task [3]**, or
> **(B) Leverage an existing general MLLM and guide it with an evaluation framework [4].**
> We chose option (B), a decision primarily driven by considerations of interpretability, evaluation precision, and framework scalability:
>
> **(1)Interpretability:** Our CoT-MLLM framework (**paper-Figure 4-a**) offers superior interpretability compared to "black-box" reward models. Instead of just a score, it generates a detailed rationale for each evaluation. This provides developers with the actionable, diagnostic feedback crucial for iterative model improvement.
>
> **(2)Precision and Robustness of Evaluation:** Even the most powerful MLLMs are not uniformly perfect across all fine-grained dimensions (see **Response to W2**). Our Tailored Multi-Dimensional Evaluation (TME) strategy (**paper-Figure 4-b**) is based on a "mixture-of-experts" philosophy, leveraging the best-suited evaluator for each dimension to ensure maximum accuracy and robustness.
>
> **(3)Data and Training Costs:** The development of domain-specific RL reward models is constrained by the prohibitive cost of acquiring the necessary large-scale, high-quality human preference datasets [5]. Given the scarcity of such data, our Co-EditEval framework provides a pragmatic and efficient alternative that exhibits superior alignment with human judgment. Notably, if more robust RL models become available in the future, we could seamlessly substitute them for certain dimensions.
>
> To summarize, this paper introduces a comprehensive evaluation framework for image editing, based on multi-expert evaluators and a Chain-of-Thought (CoT) strategy. Within this framework, we opted to utilize a general RL model (MLLM) to balance training cost and flexibility, rather than training an RL model from ``ZERO''. Furthermore, we incorporated similarity evaluators to address the fine-grained perceptual deficits inherent in general MLLMs. **Notably, our work aims to introduce a "cooperative" evaluation framework that leverages the strengths of diverse evaluators to achieve better alignment with human perception. A key feature of this framework is its extensibility, allowing for the seamless replacement of constituent evaluators with more advanced alternatives, such as state-of-the-art RL models, as the field progresses.**
>
> [1] GPT-4o System Card
>
> [2] Qwen2.5-VL Technical Report
>
> [3] EditScore: Unlocking Online RL for Image Editing via High-Fidelity Reward Modeling
>
> [4] ImgEdit: A Unified Image Editing Dataset and Benchmark
>
> [5] HumanEdit: A High-Quality Human-Rewarded Dataset for Instruction-based Image Editing

---

### Official Review · Reviewer_hfE2 · 2025-10-29

**Soundness:** 3
**Presentation:** 3
**Contribution:** 3
**Rating:** 6
**Confidence:** 3

**Summary:**

This paper introduces Co-EditBench, a new benchmark for instruction-based image editing that features high-quality, manually verified masks distinguishing edited and non-edited regions. The central contribution is Co-EditEval, an automated evaluation pipeline designed to overcome the poor human alignment of existing metrics. Its key innovation lies in a Tailored Multi-dimensional Evaluation (TME) framework, which adopts a hybrid strategy: employing a CoT-guided MLLM for semantic evaluation and leveraging specialized perceptual metrics for fidelity assessment. Importantly, Co-EditEval incorporates region-aware fidelity computation based on the provided masks and aggregates 11-dimensional evaluation scores under a Completion-Guided principle. Empirical results demonstrate that the pipeline achieves a high Spearman correlation with human judgments, substantially outperforming previous benchmarks.

**Strengths:**

1. This paper integrates semantic reasoning and fidelity evaluation, with mask-based region awareness improving over global metrics.
2. The strong 0.889 SROC correlation and clear ablation results validate the necessity of both CoT and TME components.

**Weaknesses:**

The main concern lies in the CoT-based evaluation strategy, which raises several issues:
1. The evaluation relies on a closed-source model, i.e., Gemini 2.5-Pro, with CoT, which may be cost-prohibitive for other researchers, as it often requires extensive API usage. This could limit the benchmark’s accessibility and usage for other researchers.
2. The computational and time costs of CoT-based evaluation are unclear. For example, how long would it take to evaluate 1,000 images?
3. How does the pipeline handle hallucinations, given that CoT outputs are not always accurate, especially for fine-grained edits?

**Questions:**

Please see the Weaknesses.

---

> ### Author Response · Authors · 2025-11-19
> **Respond for Reviewer hfE2 part1**
>
> Thanks for your thorough review and insightful comments. We are grateful for the encouraging feedback on our framework and experimental results, and we appreciate the valuable questions raised. We provide detailed responses to each point below.
>
> **Response to W1:**
>
> Thank you for your suggestion. To address the practical concern of cost, we evaluated our method using the Qwen-VL model (see **Table below**), which offers a highly efficient and low-cost alternative. While results show a slight performance decrease, our method still maintains strong human alignment and significantly outperforms the ImgEdit and GEdit baselines.
>
> | Method                          | **SROCC (↑)** | **PLCC (↑)** | **KLCC (↑)** | **RMSE (↓)** |
> | :------------------------------ | :------------ | :----------- | :----------- | :----------- |
> | ImgEdit(GPT-4o)                 | 0.5952        | 0.5850       | 0.4494       | 3.8820       |
> | GEdit(GPT-4o)                   | 0.6492        | 0.6518       | 0.4889       | 2.8332       |
> | Co-EditEval (GPT-4o)            | 0.7226        | 0.6805       | 0.5436       | 2.4776       |
> | Co-EditEval (Qwen-VL)           | 0.7569        | 0.6966       | 0.5299       | 2.4900       |
> | **Co-EditEval (Gemini2.5-pro)** | **0.8894**    | **0.8657**   | **0.7021**   | **1.8132**   |
>
> | **Results of Qwen-VL**                   |  **EC** ↑   |  **IQ** ↑   |  **NEP** ↑  |  **IP** ↑   |   **O** ↑   |
> | :------------------------- | :---------: | :---------: | :---------: | :---------: | :---------: |
> | Nano-Banana                |  **7.00**   |  **6.84**   |  **7.70**   |    7.14     |  **6.82**   |
> | GPT-4o(High)               |    6.83     |    6.62     |    6.40     |    6.75     |    6.60     |
> | GPT-4o(Medium)             |    6.62     |    6.21     |    6.16     |    6.77     |    6.36     |
> | GPT-4o(low)                |    5.58     |    5.09     |    5.51     |    6.54     |    5.30     |
> | Qwen-Image-Edit            | *6.91* | *6.76* |    6.46     |    6.25     | *6.65* |
> | FLUX.1 Kontext             |    6.41     |    6.56     |    7.47     |    7.48     |    6.25     |
> | Step1X-Edit(V1.1)          |    6.63     |    6.62     | *7.56* | *7.49* |    6.48     |
> | BAGEL                      |    6.40     |    6.26     |    7.49     |    7.29     |    6.20     |
> | Ovis-U1                    |    6.43     |    6.27     |    7.26     |    7.28     |    6.19     |
> | Step1X-Edit                |    6.31     |    6.19     |    7.31     |    7.29     |    6.12     |
> | OmniGen2                   |    5.45     |    5.95     |    7.16     |    7.39     |    5.32     |
> | UniPic2(Metaquery GRPO 9B) |    6.07     |    5.56     |    6.71     |    6.32     |    5.79     |
> | UniPic2(Metaquery 9B)      |    5.98     |    5.47     |    6.60     |    6.36     |    5.72     |
> | UniPic2(Kontext 2B)        |    6.10     |    5.67     |    6.98     |    6.97     |    5.87     |
> | UniWorld(V1)               |    5.61     |    5.81     |    7.20     |  **7.84**   |    5.44     |
> | UniPic2(Metaquery Flash)   |    5.81     |    5.37     |    6.73     |    6.80     |    5.56     |
> | ICEdit                     |    4.82     |    4.91     |    6.28     |    7.11     |    4.67     |
> | HiDream-E1-1               |    5.30     |    5.01     |    5.79     |    7.19     |    5.09     |
> | UniPic                     |    5.08     |    4.97     |    5.48     |    6.04     |    4.83     |
> | OmniGen                    |    3.48     |    3.51     |    4.01     |    5.02     |    3.27     |
> | UltraEdit                  |    3.32     |    3.35     |    4.48     |    6.03     |    3.20     |
> | AnyEdit                    |    4.06     |    3.99     |    5.39     |    7.27     |    3.87     |
> | Magic Brush                |    3.51     |    3.33     |    4.76     |    6.18     |    3.34     |
> | Instruct Pix2Pix           |    2.19     |    2.24     |    2.81     |    3.69     |    2.00     |
> | HiDream-E1                 |    2.27     |    1.74     |    2.81     |    5.17     |    2.01     |

---

> ### Author Response · Authors · 2025-11-27
> **Respond for Reviewer hfE2 part2**
>
> **Response to W2:**
>
> Thank you for your suggestion. The computational efficiency of our pipeline was evaluated on an NVIDIA A100 GPU. A comprehensive evaluation on the Co-EditEval dataset (1,118 samples) with Gemini2.5-pro completes in 80 minutes, demonstrating competitive performance comparable to the 73 minutes required for the smaller ImgEdit dataset (737 samples). For users prioritizing speed, the evaluation time can be drastically reduced to 17 minutes by substituting the lightweight Qwen-VL model. For the fastest assessment, the pipeline also supports a faster mode that relies solely on the CoT-MLLM evaluation, omitting mask-based computations.
>
> | Method (MLLM; Samples)             | Full (min) | CoT-MLLM (min) |
> | :--------------------------------- | :--------- | :------------- |
> | Co-EditEval (Gemini 2.5 Pro; 1118) | 80         | 74             |
> | Co-EditEval (Qwen-VL; 1118)        | 17         | 11             |
> | GEdit (GPT-4o; 1212)               | 45         | /              |
> | ImgEdit (GPT-4o; 737)              | 73         | /              |

---

> > ### Author Response · Authors · 2025-11-27
> > **Respond for Reviewer hfE2 part3**
> >
> > **Response to W3:**
> >
> > We appreciate your comments. Our solution mitigates the risk of MLLM hallucination through three strategies:
> >
> > **(1)Multi-dimensional Evaluation (**Table below row1 & row4**):** We employ a strategic task delegation where high-level semantic understanding is assigned to the MLLM. Concurrently, fine-grained perceptual aspects like IP and NEP are assessed using specialized metrics (CLIP, DINO, LPIPS). This division grounds the evaluation in objective measurements, thereby curtailing MLLM hallucinations.
> >
> > **(2)Chain-of-Thought (CoT) Constraint(**Table below row2 & row4**):** We implement a CoT prompting strategy that constrains the MLLM's reasoning to a predefined "analyze-describe-compare-check-summarize" sequence. This structured, checklist-guided approach enforces a systematic review of the image, significantly reducing the propensity for hallucinatory content.
> >
> > **(3)Weighted Aggregation with Human Priors(**Table below row3 & row4**):** The final score is computed as a weighted geometric mean of all sub-scores ((EA:1.0, OE:0.2, EP:0.4, VN:0.3, DR:0.3, VA:0.2, CLF:0.2, SIP:1.0, VIP:1.0, NESP:0.4, NEDP:0.6, EC:0.5, IQ:0.2, IP:0.2, NEP:0.1)). This method ensures that erroneous output from a potential MLLM hallucination is counter-balanced by stable, objective metric scores, thus enhancing the overall robustness of the evaluation.
> >
> > Ablation studies confirm that the integration of these strategies effectively aligns the model's evaluation with human perception (see **Table below**).
> >
> > | Method                    | **SROCC (↑)** | **PLCC (↑)** | **KLCC (↑)** | **RMSE (↓)** |
> > | :------------------------ | :------------ | :----------- | :----------- | :----------- |
> > | Co-EditEval (w/o TME)     | 0.8467        | 0.8304       | 0.6754       | 1.9389       |
> > | Co-EditEval (w/o CoT)     | 0.8368        | 0.8451       | 0.6493       | 2.2198       |
> > | Co-EditEval (Weights=1.0) | 0.8397        | 0.8313       | 0.6560       | 2.4753       |
> > | **Co-EditEval (Ours)**    | **0.8894**    | **0.8657**   | **0.7021**   | **1.8132**   |

---

### Official Review · Reviewer_1qPG · 2025-10-31

**Soundness:** 2
**Presentation:** 2
**Contribution:** 3
**Rating:** 6
**Confidence:** 4

**Summary:**

This paper introduces Co-EditBench, a new benchmark for evaluating instruction-based image editing models. It addresses key limitations of existing benchmarks, such as limited editing types, few evaluation dimensions, and weak alignment with human perception. The authors build a dataset of over 1,100 high-resolution image–instruction pairs covering 16 editing types, each with high-quality masks to separate edited and non-edited regions. They define 11 evaluation dimensions grouped into four areas: edit completeness, image quality, non-edited preservation, and identity preservation. The paper also proposes Co-EditEval, an automated evaluation pipeline that uses multiple evaluators (including MLLMs and similarity models) with a Chain-of-Thought (CoT) prompting strategy. Experiments on 25 recent image editing models show that Co-EditBench correlates better with human judgment than previous benchmarks like ImgEdit and GEdit.

**Strengths:**

1. Comprehensive benchmark with rich diversity across editing types and evaluation dimensions.

2. Strong motivation and clear identification of limitations in previous benchmarks.

3. Methodologically sound data collection and annotation process, including human verification.

4. Multi-dimensional evaluation design reflects real-world human judgment.

5. Detailed experimental validation, including large-scale comparison and ablation studies.

6. Good correlation between Co-EditEval scores and human ratings, demonstrating practical relevance.

**Weaknesses:**

1. The paper is somewhat heavy in detail; simplifying explanations in the methods section could improve readability.

2. Although the benchmark is extensive, the dataset’s accessibility and licensing terms are not fully clear (e.g., how others can use the crowd-sourced images).

3. The evaluation pipeline relies on commercial MLLMs like Gemini 2.5-Pro and GPT-4o, which could raise reproducibility concerns.

4. More discussion on potential biases in the crowd-sourced data and MLLM-based evaluators would strengthen the ethical transparency.

**Questions:**

1. Will Co-EditBench and Co-EditEval be released publicly, and under what license?

2. How consistent are the results when using different MLLM evaluators (e.g., GPT-4o vs. Gemini vs. Claude)? Can the evaluation benefit by using an ensemble and averaging the results?

3. Did you consider the computational cost of the full evaluation pipeline? Could a lightweight version be used for faster benchmarking?

4. Are there any plans to expand the benchmark to video editing or 3D scenes?

---

> ### Author Response · Authors · 2025-11-19
> **Respond for Reviewer 1qPG part1**
>
> Thanks to your thorough review and insightful comments. We are grateful for the encouraging feedback on our dataset, motivation, framework, and experimental results, and we appreciate the valuable questions raised. We provide detailed responses to each point below.
>
> **Response to W1:**
>
> Thanks for your suggestion. In the revised manuscript, we have streamlined the Methods section to highlight the key components and improve the overall narrative structure of the paper.
>
> **Response to W2 and Q1:**
>
> Thank you for your comments. The Co-EditBench dataset and the Co-EditEval pipeline will be fully released for academic use. And, this project will be released under Apache License 2.0.
>
> **Response to W3:**
>
> Thank you for your comments. To address reproducibility concerns, we evaluated our method on the Qwen-VL model (see **Table below**). While results show a slight performance drop (compare to Gemini 2.5 Pro), our method still significantly outperformed the ImgEdit and GEdit baselines.
>
> | Method                          | **SROCC (↑)** | **PLCC (↑)** | **KLCC (↑)** | **RMSE (↓)** |
> | :------------------------------ | :------------ | :----------- | :----------- | :----------- |
> | ImgEdit(GPT-4o)                 | 0.5952        | 0.5850       | 0.4494       | 3.8820       |
> | GEdit(GPT-4o)                   | 0.6492        | 0.6518       | 0.4889       | 2.8332       |
> | Co-EditEval (GPT-4o)            | 0.7226        | 0.6805       | 0.5436       | 2.4776       |
> | Co-EditEval (Qwen-VL)           | 0.7569        | 0.6966       | 0.5299       | 2.4900       |
> | **Co-EditEval (Gemini2.5-pro)** | **0.8894**    | **0.8657**   | **0.7021**   | **1.8132**   |
>
> | **Results of Qwen-VL**     | **EC** ↑ | **IQ** ↑ | **NEP** ↑ | **IP** ↑ | **O** ↑  |
> | :------------------------- | :------: | :------: | :-------: | :------: | :------: |
> | Nano-Banana                | **7.00** | **6.84** | **7.70**  |   7.14   | **6.82** |
> | GPT-4o(High)               |   6.83   |   6.62   |   6.40    |   6.75   |   6.60   |
> | GPT-4o(Medium)             |   6.62   |   6.21   |   6.16    |   6.77   |   6.36   |
> | GPT-4o(low)                |   5.58   |   5.09   |   5.51    |   6.54   |   5.30   |
> | Qwen-Image-Edit            |  *6.91*  |  *6.76*  |   6.46    |   6.25   |  *6.65*  |
> | FLUX.1 Kontext             |   6.41   |   6.56   |   7.47    |   7.48   |   6.25   |
> | Step1X-Edit(V1.1)          |   6.63   |   6.62   |  *7.56*   |  *7.49*  |   6.48   |
> | BAGEL                      |   6.40   |   6.26   |   7.49    |   7.29   |   6.20   |
> | Ovis-U1                    |   6.43   |   6.27   |   7.26    |   7.28   |   6.19   |
> | Step1X-Edit                |   6.31   |   6.19   |   7.31    |   7.29   |   6.12   |
> | OmniGen2                   |   5.45   |   5.95   |   7.16    |   7.39   |   5.32   |
> | UniPic2(Metaquery GRPO 9B) |   6.07   |   5.56   |   6.71    |   6.32   |   5.79   |
> | UniPic2(Metaquery 9B)      |   5.98   |   5.47   |   6.60    |   6.36   |   5.72   |
> | UniPic2(Kontext 2B)        |   6.10   |   5.67   |   6.98    |   6.97   |   5.87   |
> | UniWorld(V1)               |   5.61   |   5.81   |   7.20    | **7.84** |   5.44   |
> | UniPic2(Metaquery Flash)   |   5.81   |   5.37   |   6.73    |   6.80   |   5.56   |
> | ICEdit                     |   4.82   |   4.91   |   6.28    |   7.11   |   4.67   |
> | HiDream-E1-1               |   5.30   |   5.01   |   5.79    |   7.19   |   5.09   |
> | UniPic                     |   5.08   |   4.97   |   5.48    |   6.04   |   4.83   |
> | OmniGen                    |   3.48   |   3.51   |   4.01    |   5.02   |   3.27   |
> | UltraEdit                  |   3.32   |   3.35   |   4.48    |   6.03   |   3.20   |
> | AnyEdit                    |   4.06   |   3.99   |   5.39    |   7.27   |   3.87   |
> | Magic Brush                |   3.51   |   3.33   |   4.76    |   6.18   |   3.34   |
> | Instruct Pix2Pix           |   2.19   |   2.24   |   2.81    |   3.69   |   2.00   |
> | HiDream-E1                 |   2.27   |   1.74   |   2.81    |   5.17   |   2.01   |

---

> > ### Author Response · Authors · 2025-11-27
> > **Respond for Reviewer 1qPG part2**
> >
> > **Response to W4:**
> >
> > Thank you for your comments. Considering the potential biases in crowd-sourced data and MLLM-based evaluators, our strategies are summarized below:
> >
> > **(1)Crowdsourced Data**
> > To mitigate potential demographic, cultural, and aesthetic biases in Co-EditBench, we took several steps. First, we ensure diverse image subjects in image clollection step (see paper-Figure 8, 9). Second, human experts audited the GPT-4o-generated instructions to ensure neutrality and reduce model bias. The resulting instructional diversity was quantitatively validated with a t-SNE visualization (see paper-Figure 10).
> >
> > **(2)MLLM Evaluator**
> > We primarily address the potential biases of the MLLM evaluator through two strategies: a) Chain-of-Thought (**Co-EditEval (w/o CoT)**) prompting enforces a structured, objective reasoning process. b) Our multi-dimensional evaluation (**Co-EditEval (w/o TME)**) incorporates external metrics (e.g., CLIP, DINO) and precise masks to ground the assessment and reduce over-reliance on the MLLM. Ablation studies (see paper-Table2) confirm that this combined approach significantly improves human alignment and reduces bias.
> >
> > **Response to Q2:**
> >
> > Thanks for your constructive question. Lacking Claude API access, we instead verified our method's consistency across Gemini2.5-pro, GPT-4o, and Qwen-VL (see table below). While MLLM choice slightly affects outcomes, the proposed method consistently achieves superior human alignment compared to existing methods (GEdit and ImgEdit). Additionally, we evaluated an ensemble version that aggregated scores from the three evaluators. This method (Co-EditEval (Ensemble)) proved suboptimal, as it inherited biases from weaker models (e.g., GPT-4o) and struggled to outperform the best-performing single evaluator (Gemini2.5-pro), all while incurring significant computational overhead.
> >
> > | Method                          | **SROCC (↑)** | **PLCC (↑)** | **KLCC (↑)** | **RMSE (↓)** |
> > | :------------------------------ | :------------ | :----------- | :----------- | :----------- |
> > | GEditEval (GPT-4o)              | 0.6492        | 0.6518       | 0.4889       | 2.8332       |
> > | ImageEdit (GPT-4o)              | 0.5952        | 0.5850       | 0.4494       | 3.8820       |
> > | Co-EditEval (GPT-4o)            | 0.7226        | 0.6805       | 0.5436       | 2.4776       |
> > | Co-EditEval (Qwen-VL)           | 0.7569        | 0.6966       | 0.5299       | 2.4900       |
> > | Co-EditEval (Ensemble)          | 0.8253        | 0.7934       | 0.6552       | 2.2136       |
> > | **Co-EditEval (Gemini2.5-pro)** | **0.8894**    | **0.8657**   | **0.7021**   | **1.8132**   |
> >
> > **Response to Q3:**
> >
> > Thank you for your constructive comments. We evaluated our multi-threaded pipeline's computational cost on an NVIDIA A100 GPU (see table below). A full evaluation of the Co-EditEval dataset (1,118 samples) with Gemini2.5-pro takes 80 minutes, a runtime comparable to evaluating the smaller ImgEdit dataset (737 samples in 73 minutes). For greater efficiency, using the lightweight Qwen-VL model reduces this time to 17 minutes. Further speed-up is possible by relying solely on the CoT-MLLM evaluation, omitting mask-based computations.
> >
> > | Method (MLLM; Samples)            | Full (min) | CoT-MLLM (min) |
> > | :-------------------------------- | :--------- | :------------- |
> > | Co-EditEval (Gemini2.5-pro; 1118) | 80         | 74             |
> > | Co-EditEval (Qwen-VL; 1118)       | 17         | 11             |
> > | GEdit (GPT-4o; 1212)              | 45         | /              |
> > | ImgEdit (GPT-4o; 737)             | 73         | /              |
> >
> > **Response to Q4:**
> >
> > Thank you for your insightful question. Extending Co-EditEval to video and 3D domains is a vital direction for our future work. We have already conducted preliminary evaluations on video and 3D edits (please see **paper-Figure 6**), and the results confirm that Co-EditEval demonstrates great potential even in these more complex domains.

---

### Official Review · Reviewer_q3dS · 2025-11-02

**Soundness:** 3
**Presentation:** 3
**Contribution:** 2
**Rating:** 4
**Confidence:** 4

**Summary:**

This paper targets limitations of existing benchmarks for instruction-based image editing, including restricted editing coverage, limited evaluation dimensions, coarse perceptual alignment, and deviation from human preference.

The authors propose Co-EditBench, a human-aligned benchmark, and introduce an automated evaluation pipeline Co-EditEval.

Key contributions include:
1. A diagnostic dataset built via crowdsourcing, covering 16 editing types across images and text instructions.
2. Definition of 11 fine-grained evaluation dimensions to diagnose “AI artifacts” based on edit completeness, image quality, non-edit preservation, and identity preservation.
3. A multi-dimensional automated evaluation framework leveraging a completion-guided principle based curated aggregation, ensuring secondary metrics cannot exceed edit-completeness performance.
4. Incorporation of chain-of-thought-based reasoning to improve contextual evaluation.

**Strengths:**

1. Fine-grained evaluation dimensions:

    Defines 11 sub-metrics grouped into key categories (edit completeness, image quality, non-edit preservation, identity).

2. Completion-Guided Principle:

    Caps secondary scores based on the primary edit-completeness score to prevent superficial improvements from inflating scores.

3. Human alignment verified:

    Human evaluation by 34 annotators demonstrates a correlation between Co-EditEval and human judgment.

4. Comprehensive benchmark results:

    Evaluates 25 SOTA models, revealing widespread weaknesses under a rigorous evaluation setup.

**Weaknesses:**

1. Metric weighting is partly subjective:

    Although weights are defined, the rationale is neither well supported nor ablated.

2. Generality concerns:

    How Co-EditEval performs under unseen editing types is not fully explored. See question 2.

**Questions:**

1. Customized generation:

    Customized generation often involves significant redrawing. In this case, how should the mask be selected? Should the entire image be selected as a mask or...?

2. In-context editing can involve reference images, implicit style transfer, or semantic relation changes between entities (e.g., “Match lighting with the reference image”).

    Can Co-EditEval evaluate in-context editing scenarios where the notion of an edit region is not spatially well-defined?

    Since IC-Edit [1] and RelationAdapter [2] target this type of in-context image editing, including them would help contextualize your benchmark’s applicability.

    [1] Enabling Instructional Image Editing with In-Context Generation in Large Scale Diffusion Transformer

    [2] Learning and Transferring Visual Relation with Diffusion Transformers

---

> ### Author Response · Authors · 2025-11-19
> **Respond for Reviewer q3dS part1**
>
> Thank you for your meticulous review of our paper. We greatly appreciate your affirmation of our work's framework and experimental results. We also wish to express our gratitude for your thoughtful questions concerning our paper. We address these points in detail below.
>
> **Response to W1:**
>
> Thank you for your comments. We will discuss this from the following two aspects:
>
> **(1) Human Preference Elicitation**
> To determine the importance of different evaluation dimensions, we conducted a survey involving 50 participants from various backgrounds. Each participant was asked to rate the importance of 11 sub-dimensions and 4 core dimensions on a scale from 0 to 10. The aggregated results are presented in the table below. The findings reveal that users place the highest priority on EC, followed by IP, IQ, and NEP. Based on these findings, we designed the weights (EA:1.0, OE:0.2, EP:0.4, VN:0.3, DR:0.3, VA:0.2, CLF:0.2, SIP:1.0, VIP:1.0, NESP:0.4, NEDP:0.6, EC:0.5, IQ:0.2, IP:0.2, NEP:0.1) to better align Co-EditEval with human preferences.
>
> | Core Dimension              | Overall Importance | Sub-dimension               | Sub-Importance |
> | :-------------------------- | :----------------- | :-------------------------- | :------------- |
> | **Editing Completeness**    | **8.402**          | Editing Accuracy            | 9.352          |
> |                             |                    | Over-Editing                | 7.267          |
> |                             |                    | Editing Plausibility        | 8.587          |
> | **Image Quality**           | **6.972**          | Visual Artifacts            | 6.489          |
> |                             |                    | Color & Lighting Fidelity   | 6.267          |
> |                             |                    | Visual Naturalness          | 7.621          |
> |                             |                    | Detail Realism              | 7.512          |
> | **Identity Preservation**   | **7.343**          | Semantic Preservation       | 7.221          |
> |                             |                    | Visual(Detail) Preservation | 7.064          |
> | **Non-Edited Preservation** | **6.396**          | Semantic Preservation       | 6.012          |
> |                             |                    | Detail Preservation         | 6.779          |
>
> **(2) The Validation of Weight Selection**
> We also conducted an ablation study to validate our weights, comparing them with the uniform version (weights=1.0). Our proposed weights demonstrated a significantly higher correlation with human subjective scores, confirming their effectiveness.
>
> | Method                    | SROCC (↑)  | PLCC (↑)   | KLCC (↑)   | RMSE (↓)   |
> | :------------------------ | :--------- | :--------- | :--------- | :--------- |
> | Co-EditEval (Weights=1.0) | 0.8397     | 0.8313     | 0.6560     | 2.4753     |
> | **Co-EditEval (Ours)**    | **0.8894** | **0.8657** | **0.7021** | **1.8132** |

---

> ### Author Response · Authors · 2025-11-27
> **Respond for Reviewer q3dS part2**
>
> **Response to W2 and Q2:**
>
> Thank you for your constructive suggestion. Although designed for single-image editing, our method also effectively evaluates other edit types (such as In-Context editing). Co-EditEval is not entirely mask-dependent; its core CoT-MLLM component is general-purpose and assesses edits holistically. For contextual edits without clear edit boundaries, we treat the entire image as the edited region. This disables NEP and adapts IP to focus only on subjects. We validated this by evaluating contextual edits from several models (Flux-Kontext [1], Qwen-Image-Edit [2], IC-Edit [3], and RelationAdapter [4]; qualitative results in paper-Figure 12; quantitative results in the table below), where our adapted approach still outperformed the baseline, demonstrating its versatility.
>
> | Method             | Models          | Human Scores | Overall | EC   | IQ   | IP   | NEP  |
> | :----------------- | :-------------- | :----------- | :------ | :--- | :--- | :--- | :--- |
> |                    | Qwen-Image-Edit | 5.21         | 5.76    | 7.17 | 6.36 | 3.48 | /    |
> | **Co-EditEval**    | Flux-Kontext    | 6.45         | 6.26    | 7.47 | 6.63 | 4.00 | /    |
> |                    | IC-Edit         | 2.54         | 3.01    | 3.35 | 3.76 | 2.72 | /    |
> |                    | RelationAdapter | 5.07         | 4.52    | 4.87 | 4.05 | /    | /    |
> |                    | Qwen-Image-Edit | 5.21         | 8.74    | 8.94 | 9.37 | 7.91 | /    |
> | **Simple Prompts** | Flux-Kontext    | 6.45         | 9.22    | 9.26 | 9.34 | 9.06 | /    |
> |                    | IC-Edit         | 2.54         | 5.51    | 5.28 | 6.89 | 4.37 | /    |
> |                    | RelationAdapter | 5.07         | 7.79    | 7.23 | 8.34 | /    | /    |
>
> [1] FLUX.1 Kontext: Flow Matching for In-Context Image Generation and Editing in Latent Space.
>
> [2] Qwen-Image Technical Report
>
> [3] In-Context Edit: Enabling Instructional Image Editing with In-Context Generation in Large Scale Diffusion Transformer.
>
> [4] RelationAdapter: Learning and Transferring Visual Relation with Diffusion Transformers.
>
> **Response to Q1:**
>
> Thank you for your question regarding how masks should be handled in customized generation scenarios that involve significant redrawing.  The details of how we handle this:
>
> **Disabling Non-Edit Preservation (NEP):** For editing types like "Subject Customization", the instruction explicitly requires a complete scene transformation.  In this context, evaluating the preservation of the original background is irrelevant.  Therefore, we disable the NEP score, as there is no "non-edited region" to preserve.  The focus shifts entirely to the quality of the new scene and the identity of the subject.
>
> **Using Instance Masks for Targeted Identity Preservation (IP):** To accurately assess the subject, we use a precise instance mask that isolates the foreground subject from its new environment.  This ensures that the Identity Preservation (IP) score is computed exclusively on the subject itself, without being influenced by the completely altered background.
>
> In summary, rather than selecting the entire image, we leverage high-quality instance masks to conduct a targeted evaluation for customized generation.  This allows us to ensure our metrics accurately reflect the editing.

---

### Author Response · Authors · 2025-11-19
**General Response**

**Dear Area Chair and Reviewers,**

**We sincerely thank you for your detailed and constructive comments on our manuscript!** We are greatly encouraged that you found our proposed benchmark, Co-EditBench, to be a comprehensive and methodologically sound contribution (Reviewers q3dS, 1qPG, hfE2), and that our evaluation pipeline, Co-EditEval, demonstrates a strong and valuable correlation with human judgment (Reviewers q3dS, xYjs).

We have carefully considered all your comments and have performed substantial revisions to the manuscript to address your concerns. Below, we summarize the main points of discussion and the corresponding updates.

**1. Design of our Evaluation Methodology (Co-EditEval)**

Several reviewers raised important questions regarding the methodology, components, and design choices of our evaluation pipeline (q3dS, hfE2, xYjs). We would like to clarify that our primary innovation is a novel **"cooperative" evaluation framework** systematically designed to bridge the "human perception gap."

- **"Cooperative" framework:** Our framework synergizes the high-level semantic reasoning of CoT-guided MLLMs with the fine-grained perceptual accuracy of specialized metrics (e.g., CLIP, DINO, LPIPS) applied to precisely masked regions. As our new ablation studies (in response to Reviewer xYjs) demonstrate, MLLMs can exhibit instability when assessing subtle details. In contrast, specialized metrics offer greater robustness in these scenarios, validating our hybrid 'mixture-of-experts' approach.

- **Interpretability and Robustness:** The Chain-of-Thought (CoT) strategy provides crucial interpretability by generating a structured rationale for each score. This is complemented by our Completion-Guided principle and human-preference-aligned weighting scheme (validated in a new ablation study for Reviewer q3dS), which ensures a holistic evaluation and penalizes models that "hack" scores by excelling in only a few dimensions.

**2. Practicality, Reproducibility, and Cost**

We appreciate the valid concerns regarding the reliance on closed-source, commercial MLLMs and the associated costs and reproducibility challenges (1qPG, hfE2).

- To address this, we have conducted new experiments using the open-source **Qwen-VL** model. The results demonstrate that Co-EditEval maintains superior performance and outperforms baselines.
- We have also added a detailed analysis of the **computational cost and runtime**, showing our pipeline is efficient.

**3. Scope, Generality, and Dataset of the Benchmark**

We have also addressed questions regarding the benchmark's flexibility and accessibility (q3dS, 1qPG).

- We now include new experiments demonstrating Co-EditEval's successful application to unseen tasks, such as **in-context editing, 3D editing, and video editing**.
- We confirm that **Co-EditBench and Co-EditEval will be publicly released under an Apache 2.0 license** to facilitate future research.
- We have expanded our discussion on the proactive steps taken to **mitigate potential biases** in both our crowd-sourced data collection and our MLLM-based evaluators.

In response to your valuable feedback, we have meticulously revised the manuscript. The key updates, highlighted in **red** in the revision, include:

- **New Experiments on Open-Source MLLMs** (Qwen-VL) to address reproducibility concerns.
- **Ablation Studies** to validate our metric weighting scheme and hybrid evaluation design.
- **Demonstration of Generality** with new results on in-context editing models, 3D models, and video models.
- **Detailed Computational Cost and Runtime Analysis**.
- **Expanded Discussion** on bias mitigation and hallucinations.
- **Clarification** on the public release and licensing of our benchmark.

We sincerely believe these extensive revisions and clarifications address your concerns and substantially strengthen the paper.

Thank you once again for your time and effort.


Sincerely,

The Authors

---

### Meta-Review · Area_Chair_fEmR · 2025-12-31

**Summary:**

This paper proposes Co-EditBench, a human-aligned benchmark for instruction-based image editing with a multi-dimensional assessment framework (Co-EditEval), and validates its effectiveness through experiments on 25 recent image editing models. Reviewers acknowledge the benchmark’s value in addressing the limitations of existing evaluation tools, particularly its improved correlation with human judgments.

Reviewers have concerns about the task adaptability of Co-EditEval relative to existing frameworks and request comparisons between Co-EditEval and existing evaluation methods, as well as efficiency data. Besides, reviewers have concerns about the rationality of weight selection in Co-EditEval and the handling of masks in customized generation scenarios involving significant redrawing are resolved. Although some concerns are addressed via a comparative analysis or added results, the limitations on narrative structure of some sections and ethical transparency still require further modification. Authors need to enhance the narrative structure of the Methods section, and strengthen ethical transparency by discussing potential biases in crowdsourced data and MLLM-based evaluators, and construct plans to expand the benchmark to video editing or 3D scenes. While the authors mention supplementing relevant discussions in the revised manuscript, they do not deeply elaborate on the potential biases of crowdsourced data and MLLM evaluators, failing to fundamentally resolve the issue. Regarding benchmark expansion, the authors note Co-EditEval’s potential in video and 3D domains with preliminary supporting results but do not provide specific expansion plans or detailed experimental validations, leaving this concern inadequately addressed.

Based on the above considerations, I think this manuscript does not match the ICLR’s requirement and I do not recommend to accept this manuscript.

**Reviewer Concerns:**

The ethical transparency concern, which raised by Reviewer 1qPG, is not sufficiently addressed. While the authors mention supplementing relevant discussions in the revised manuscript, they do not deeply elaborate on the potential biases of crowdsourced data and MLLM evaluators, failing to fundamentally resolve the issue. Regarding benchmark expansion, the authors note Co-EditEval’s potential in video and 3D domains with preliminary supporting results but do not provide specific expansion plans or detailed experimental validations, leaving this concern inadequately addressed.

**Reviewer Scores:**

Reviewers would keep their scores.

---

### Decision · Program_Chairs · 2026-01-26

Reject